# Positional Encoding meets Persistent Homology on Graphs

Yogesh Verma [1]   Amauri H. Souza [1 2]   Vikas Garg [1 3]

## Abstract

The local inductive bias of message-passing graph neural networks (GNNs) hampers their ability to exploit key structural information (e.g., connectivity and cycles). Positional encoding (PE) and Persistent Homology (PH) have emerged as two promising approaches to mitigate this issue. PE schemes endow GNNs with location-aware features, while PH methods enhance GNNs with multiresolution topological features. However, a rigorous theoretical characterization of the relative merits and shortcomings of PE and PH has remained elusive. We bridge this gap by establishing that neither paradigm is more expressive than the other, providing novel constructions where one approach fails but the other succeeds. Our insights inform the design of a novel learnable method, PiPE (Persistence-informed Positional Encoding), which is provably more expressive than both PH and PE. PiPE demonstrates strong performance across a variety of tasks (e.g., molecule property prediction, graph classification, and out-of-distribution generalization), thereby advancing the frontiers of graph representation learning. Code is available at https://github.com/Aalto-QuML/PIPE.

## 1. Introduction

Many natural systems, such as social networks (Freeman, 2004) and proteins (Jha et al., 2022), exhibit complex relational structures often represented as graphs. To tackle prediction problems in these domains, message-passing graph neural networks (GNNs) (Scarselli et al., 2009; Bronstein et al., 2017; Hamilton et al., 2017; Velickovic et al., 2018) have become the dominant approach, leading to breakthroughs in diverse applications such as drug discovery (Gilmer et al., 2017; Stokes et al., 2020; Satorras et al., 2021), simulation of physical systems (Cranmer et al., 2019; Sanchez-Gonzalez et al., 2020; Verma & Jena, 2021), algorithmic reasoning (Dudzik et al., 2023; Jurss et al., 2023), and recommender systems (Ying et al., 2018).

Despite this success, message-passing GNNs have rather limited expressivity — they are at most as powerful as the 1-Weisfeiler-Lehman (1-WL) test (Weisfeiler & Leman, 1968) in distinguishing non-isomorphic graphs (Xu et al., 2019; Morris et al., 2019; Nikolentzos et al., 2023). This inherent limitation has prompted the development of more expressive GNNs by leveraging, e.g., topological features (Horn et al., 2022), random features (Sato et al., 2021), higher-order message passing (Morris et al., 2019; Ballester et al., 2024), and structural/positional encodings (Li et al., 2020; You et al., 2019; Wang et al., 2023).

Inspired by the success of positional encodings (PEs) in Transformers (Vaswani et al., 2017) for sequences, several positional encodings for graphs have been proposed (You et al., 2019; Dwivedi et al., 2022; Wang et al., 2023; Huang et al., 2024). For instance, spectral methods exploit global structure via the eigendecomposition of the graph Laplacian (Lim et al., 2023; Kreuzer et al., 2021; Huang et al., 2024). However, these encodings suffer from inherent ambiguities due to sign flips, basis changes, stability, and eigenvalue multiplicities. Recent efforts have addressed sign and basis symmetries (Lim et al., 2023; Wang et al., 2023) and stability with respect to graph perturbations (Huang et al., 2024). However, a common drawback persists: most methods partition the Laplacian eigenvalue/eigenvector space and utilize only the partitioned eigenvalues/eigenvectors. This approach discards valuable information contained in the remaining eigenvalues and eigenvectors. Another class of methods leverage relative distances (e.g., computed from random walk diffusion) to anchor-nodes to capture structural information (Dwivedi et al., 2022; Eliasof et al., 2023; Ying et al., 2021; You et al., 2019; Li et al., 2020). Despite these advances, existing methods fail to extract detailed multiscale topological information, such as the persistence of connected components and independent cycles (i.e., 0- and 1-dim topological invariants), which may be relevant to downstream tasks and potentially more expressive.

Persistent homology (PH) (Edelsbrunner et al., 2002) is

[1]Department of Computer Science, Aalto University, Finland [2]Federal Institute of Ceará [3]YaiYai Ltd. Correspondence to: Yogesh Verma <yogesh.verma@aalto.fi>.

*Proceedings of the 42nd International Conference on Machine Learning*, Vancouver, Canada. PMLR 267, 2025. Copyright 2025 by the author(s).

| Main contributions | |
| --- | --- |
| **Shortcomings of PE and PH (Section 3):** | |
| Neither is more expressive: constructions exposing limitations | Prop. 3.1, 3.2 |
| **PH enhanced with PE (Section 3.1):** | |
| Different base PE $\to$ different persistent diagrams | Lem. 3.3 |
| Comparison with standalone PE methods | Prop. 3.4, 3.5 |
| On degree-based filtrations and PE | Prop. 3.6 |
| **PiPE (Section 4):** | |
| LPE-based PiPE $\succ$ LPE-based LSPE, PH+LPE | Prop. 4.1, 4.2 |
| RW-based PiPE and 3-WL | Prop. 4.3 |
| On $k$-FWL and color separating sets | Prop. 4.4 |
| **Experiments (Section 5):** | |
| Graph classification, property prediction, and OOD tasks | |

Figure 1: **Overview of our key contributions**

the cornerstone of topological data analysis and offers a powerful framework to capture multi-scale topological information from data. In the context of graphs, PH has been recently used, e.g., to boost the expressive and representational power of GNNs (Horn et al., 2022; Immonen et al., 2023; Carriere et al., 2020; Verma et al., 2024). However, while both PE and PH schemes enhance the expressivity of GNNs, their relative merits and shortcomings remain unclear. Furthermore, whether the two can be harmonized to enable further expressivity gains remains unexplored.

In this work, we introduce novel constructions to reveal that neither paradigm is more expressive than the other. Leveraging our insights, we introduce **PiPE** (**P**ersistence-**i**nformed **P**ositional **E**ncoding), a *learnable* positional encoding scheme that unifies PE and PH through message-passing networks. Notably, PiPE is very flexible as it can be based on any existing PE method for graphs, and renders provable expressivity benefits over what can be achieved by either PE or PH methods on their own.

Specifically, we theoretically analyze PiPE and compare its representational power to popular learnable PE methods such as LSPE (Dwivedi et al., 2022), and analyze it in terms of the higher-order WL hierarchy, i.e., $k$-WL. To demonstrate the effectiveness of our proposal, we conduct rigorous empirical evaluations on various tasks, including molecule property prediction, out-of-distribution generalization, and synthetic tree tasks.

In sum, **our contributions** are three-fold:

1. (**Theory**) We establish theoretical results about incomparability of PE and PH methods, their limitations and how we can combine PH and PE to elevate the representational power – summarized in Figure 1.

2. (**Methodolgy**) Building on these insights, we introduce Persistence-informed Positional Encoding (PiPE), a novel *learnable* PE method that unifies PE and PH through message-passing networks by combining strengths of both.

3. (**Empirical**) We show that the improved expressivity of our approach also translates into gains in real-world problems such as graph classification, molecule property prediction, out-of-distribution generalization, as well as on synthetic tree tasks.

## 2. Background

This section overviews graph positional encoding methods, and some basic notions in persistent homology for graphs.

**Notation.** We define a graph as a tuple $G = (V, E)$, where $V = \{1, \dots, n\}$ is a set of vertices (or nodes) and $E$ is a set of unordered pairs of vertices, called edges. We denote the adjacency matrix of $G$ by $A \in \{0, 1\}^{n \times n}$, i.e., $A_{ij}$ is one if $\{i, j\} \in E$ and zero otherwise. We use $D$ to represent the diagonal degree matrix of $G$, i.e., $D_{ii} = \sum_j A_{ij}$. We define the normalized Laplacian of $G$ as $\Delta = I_n - D^{-1/2} A D^{-1/2}$ and its random walk Laplacian as $\Delta_{\text{RW}} = D^{-1} A$, where $I_n$ is the $n$-dimensional identity matrix. The set of neighbors of a node $v$ is denoted by $\mathcal{N}(v) = \{u \in V : \{v, u\} \in E\}$. Furthermore, we use $\{\!\{\cdot\}\!\}$ to denote multisets. Attributed graphs are augmented with a function $x : V \to \mathbb{R}^d$ that assigns a color (or $d$-dimensional feature vector) to nodes $v \in V$ — for notational convenience, hereafter, we denote the feature vector of $v$ by $x_v$. Finally, two attributed graphs $G = (V, E, x)$ and $G' = (V', E', x')$ are said to be *isomorphic* if there is a bijection $g : V \to V'$ such that $\{u, v\} \in E$ iff $\{g(u), g(v)\} \in E'$ and $x' \circ g = x$.

PiPE integration with backbone GNN

Readout (graph-level)

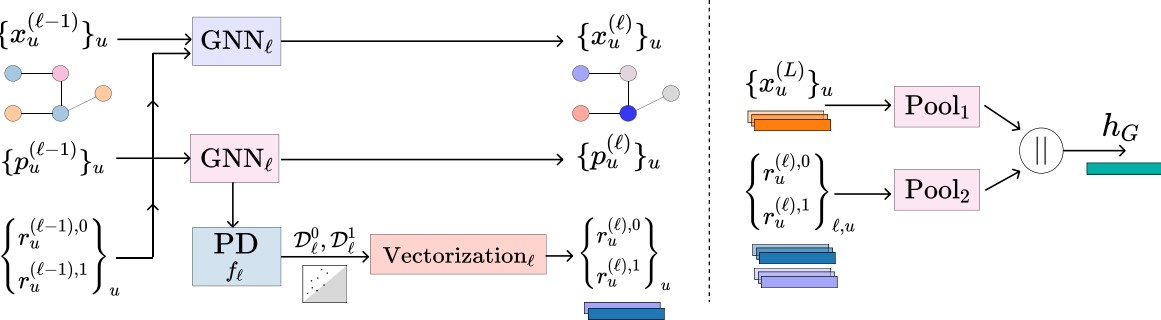

Figure 2: **Overview of PiPE and integration with backbone GNN.** At each layer $\ell$, the node embeddings $\{x_u^{\ell-1}\}_u$ are updated using the positional embeddings $\{p_u^{\ell-1}\}_u$ and the topological embeddings $\{r_u^{\ell-1,0}, r_u^{\ell-1,1}\}_u$. The position embeddings $\{p_u^{\ell-1}\}_u$ are updated and then leading to the computation of persistence diagrams $\mathcal{D}_\ell^0, \mathcal{D}_\ell^1$ leading to topological embeddings $\{r_u^{\ell,0}, r_u^{\ell,1}\}_u$. In readout phase, the final layer node embeddings $\{x_u^L\}_u$ are combined with the topological embeddings $\{r_u^{\ell,0}, r_u^{\ell,1}\}_{u,\ell}$ for various tasks.

## 2.1. Graph positional encoding

Given a graph $G$, a positional encoder acts on $A$ (adjacency matrix of $G$) to obtain an embedding matrix $P \in \mathbb{R}^{n \times k}$, where the $v$-th row of $P$ comprise the positional feature of node $v$, denoted by $p_v$. Integrating PEs into message-passing GNNs (Gilmer et al., 2017; Xu et al., 2019) enables them to learn intricate relationships between nodes based on positional information, ultimately enhancing their representational power. Although several PE methods (Dwivedi et al., 2022; Li et al., 2020; Lim et al., 2023; Wang et al., 2023) have been proposed, most approaches build upon:

- Laplacian PE (Dwivedi & Bresson, 2020): This approach employs the idea of Laplacian eigenmaps (Belkin & Niyogi, 2003) as PE. In particular, let $\Delta = U\Lambda U^\top$, where $U \in \mathbb{R}^{n \times n}$ is an orthonormal matrix with eigenvectors $u_1, \ldots, u_n$ and the matrix $\Lambda = \mathrm{diag}(\lambda_1, \ldots, \lambda_n)$ comprises the corresponding eigenvalues (or spectrum) of $\Delta$, with $\lambda_1 \leq \lambda_2 \leq \cdots \leq \lambda_n$. Then, Laplacian PE uses the $k$ smallest (non-trivial) eigenvectors as positional encodings, i.e., $p_v = [u_{1,v}, u_{2,v}, \ldots, u_{k,v}]$ for all $v \in V$. We note that this corresponds to the solution to: $\max_{P \in \mathbb{R}^{n \times k}} \mathrm{trace}(P^\top \Delta P)$ subject to $P^\top D P = I_k$.

- Distance PE (Li et al., 2020): Let $S \subseteq V$ be a target subset of vertices. Distance PE learns node features for each node $v$ based on distances from $v$ to elements in $S$ (You et al., 2019). The distances comprise either random walk probabilities or generalized PageRank scores (Li et al., 2019). Formally, using sum-pooling, Distance PE computes $p_v = \sum_{s \in S} f(d_G(v, s))$ with $d_G(v, s) = [(\Delta_{\mathrm{RW}})_{vs}, (\Delta_{\mathrm{RW}}^2)_{vs}, \ldots, (\Delta_{\mathrm{RW}}^k)_{vs}]$ or $d_G(v, s) = (\sum_{i=1}^k \gamma_i \Delta_{\mathrm{RW}}^i)_{vs}$, where $\gamma_i \in \mathbb{R}$ and $f(\cdot)$ is a multilayer perceptron.

- Random walk PE (Dwivedi et al., 2022): This approach captures node proximity through the random walk diffusion process and can be viewed as a simplified version of Distance PE. In particular, Dwivedi et al. (2022) adopt $p_v = [(\Delta_{\mathrm{RW}})_{vv}, (\Delta_{\mathrm{RW}}^2)_{vv}, \ldots, (\Delta_{\mathrm{RW}}^k)_{vv}]$.

Dwivedi et al. (2022) also propose *learnable structural and positional encodings* (LSPE) as a general framework that builds upon base positional encoders (e.g., LapPE). More specifically, the key idea of LPSE lies at decoupling positional and structural representations and learn them using message-passing layers. Formally, starting from $x_v^0 = x_v$ and $p_v^0 = p_v \; \forall v \in V$, LSPE recursively updates positional and node embeddings as

$$x_v^{\ell+1} = \mathrm{Upd}_\ell^x \left( x_v^\ell, p_v^\ell, \mathrm{Agg}_\ell^x(\{\!\{x_u^\ell, p_u^\ell : u \in \mathcal{N}(v)\}\!\}) \right) \quad (1)$$

$$p_v^{\ell+1} = \mathrm{Upd}_\ell^p \left( p_v^\ell, \mathrm{Agg}_\ell^p(\{\!\{p_u^\ell : u \in \mathcal{N}(v)\}\!\}) \right), \quad (2)$$

where $\mathrm{Agg}_\ell^p$ and $\mathrm{Agg}_\ell^x$ are arbitrary order-invariant functions, and $\mathrm{Upd}_\ell^x$ and $\mathrm{Upd}_\ell^p$ are arbitrary functions (often multilayer perceptrons, MLPs). After iterative updating, the final layer node embeddings are concatenated with the final positional ones, i.e., $\{[x_v^L, p_v^L]\}_v$, and then leveraged for downstream tasks, such as node classification, graph classification, or link prediction.

## 2.2. Persistent homology on graphs

A key notion in persistent homology is that of filtration. In this regard, a *filtration* of a graph $G$ is a finite nested sequence of subgraphs of $G$, i.e., $\emptyset = G_0 \subset G_1 \subset \ldots \subset G$. A popular choice to obtain a filtration consists of considering sublevel sets of a function defined on the vertices of a graph. In particular, let $f : V \to \mathbb{R}$ be a filtering function and $G_\alpha$ be the subgraph of $G$ induced by the vertex set $V_\alpha = \{v : f(v) \leq \alpha\}$ for $\alpha \in \mathbb{R}$. By varying $\alpha$ from

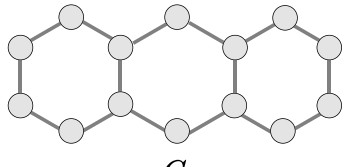 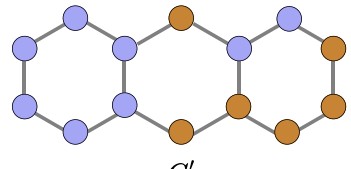 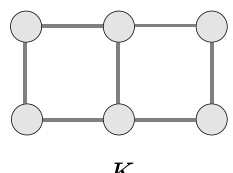 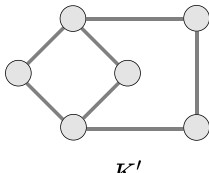

$G$ $\qquad\qquad$ $G'$ $\qquad\qquad$ $K$ $\qquad\qquad$ $K'$

Figure 3: **Incomparability of PH and PE.** The graph $G$, resembling an anthracene molecule, consists of three conjoined 3-cycles sharing a ring. Using the $k = \beta_0(G) + 1$ lowest Laplacian PE eigenmaps, PE fails to capture the number of basis cycles. The $k^{\text{th}}$ (Fiedler's) eigenvector partitions $G$ into two components, as shown in $G'$, missing the cyclic structure. The graphs $K$ consist of two 4-cycles sharing consecutive nodes, and $K'$ consist of a 4-cycle inscribed in a 5-cycle sharing three consecutive nodes. Both $K$ and $K'$ have the same number of connected components ($\beta_0$) and basis cycles ($\beta_1$) but have distinct Laplacian eigenspectra.

$-\infty$ to $\infty$, we obtain a sub-level filtration of $G$. Importantly, we can monitor the emergence and vanishing of topological characteristics (e.g., connected components, loops) throughout a filtration, which is the core idea of PH. More specifically, if a topological feature first appears in $G_{\alpha_b}$ and disappears in $G_{\alpha_d}$, then we encode its persistence as a pair $(\alpha_b, \alpha_d)$; if a feature does not disappear, then its persistence is $(\alpha_b, \infty)$. The collection of all pairs forms a multiset that we call *persistence diagram* (PD). We use $\mathcal{D}^i(G, f)$ to denote the $i$-dimensional PD of $G$ obtained using the function $f$. Additionally, for any graph $G$, $\beta_0(G)$ and $\beta_1(G)$ are its Betti numbers, i.e., the number of connected components and independent (basic) cycles, respectively. For a formal treatment of PH, we refer to Edelsbrunner & Harer (2010) and Hensel et al. (2021).

In graph learning, persistent homology has been harnessed to enhance the expressive power of GNNs. Horn et al. (2022) introduced TOGL, a general framework for integrating topological features derived from PH into GNN layers. TOGL employs a learnable function (a multilayer perceptron, MLP) on node features / colors to obtain graph filtrations, which we refer to as *vertex-color* (VC) filtrations. Importantly, Immonen et al. (2023) characterized the expressive power of VC filtrations via the notions of *color-separating sets* and *component-wise colors*. Formally, a color-separating set for a pair of attributed graphs $(G, G')$ with corresponding coloring functions $x$ and $x'$ is a set of colors $Q$ such that the subgraphs induced by $V \setminus \{w \in V \mid x_w \in Q\}$ and $V' \setminus \{w \in V' \mid x'_w \in Q\}$ have distinct component-wise colors — defined as the multiset comprising the set of node colors of each connected component. When dealing with VC filtrations, we use $\mathcal{D}^i(G, C, f)$ to denote the $i$-th dim persistence diagram of $G$ considering the indexed set of vertex colors $C = \{c_v\}_{v \in V(G)}$, with $c_v \in \mathcal{U}$, and the filtration function $f : \mathcal{U} \to \mathbb{R}$ — $\mathcal{U}$ is the universe of possible colors.

## 3. Shortcomings of PE and PH

This section examines the limitations of positional encodings (PE) and PH methods in distinguishing non-isomorphic graphs. We focus on Laplacian PE, random-walk PE, and distance encodings (see Section 2.1) as they are the fundamental building blocks behind most PE methods. Regarding PH, we mainly consider persistence diagrams derived from VC filtrations. We defer all proofs to Appendix C.

Our first result (Proposition 3.1) establishes the incomparability of PH and LapPE on unattributed graphs, i.e., neither method encompasses the capabilities of the other. Recall that the expressive power of PH from VC filtrations on unattributed graphs is bounded by $\beta_0$ and $\beta_1$.

**Proposition 3.1.** *For any graph $G$ and $k > 0$, let $\Phi_k(G)$ denote the multiset of Laplacian positional encodings (LPE) of $G$ built using the $k$ lowest eigenpairs. The following holds:*

*S1. There exist $G_1 \not\cong G_2$ s.t. $\beta_0(G_1) = \beta_0(G_2)$, $\beta_1(G_1) = \beta_1(G_2)$ but $\Phi_k(G_1) \neq \Phi_k(G_2)$ for all $k > \beta_0(G_1)$;*

*S2. There exist $G_1 \not\cong G_2$ such that $\beta_1(G_1) \neq \beta_1(G_2)$ but $\Phi_k(G_1) = \Phi_k(G_2)$;*

*S3. There exist $G_1 \not\cong G_2$ such that $\beta_0(G_1) \neq \beta_0(G_2)$ but $\Phi_k(G_1) = \Phi_k(G_2)$.*

This shows that the incomparability holds even if we consider only 0-dim or 1-dim topological features. Proposition 3.2 shows the same result also applies to random walk PE.

**Proposition 3.2.** *Let $\Phi_k(G)$ denote the multiset of random walk positional encodings obtained from walks of length $k$. The statements S1, S2, S3 of Proposition 3.1 still holds.*

Overall, Propositions 3.1 and 3.2 reveal that both PE and PH methods have complementary limitations. A key consequence of statement S2 is that LapPE cannot determine the number of basis cycles in a graph. As illustrated in Figure 3, even when using $k = \beta_0 + 1$ lowest Laplacian eigenmaps, the $k^{th}$ (Fiedler's) eigenvector merely bisects graph $G'$ into two components, failing to capture the cyclic structure.

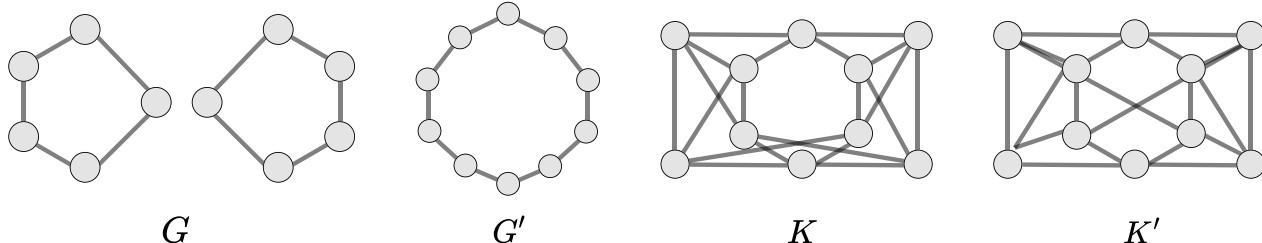

Figure 4: **Construction for Propositions 3.4 and 4.3**. The graph $G$ with two copies of $C_5$ is indistinguishable from graph $G'$ with $C_{10}$ by RW-based PE , despite having disconnected components. This follows since every node in $G$ and $G'$ has the same degree and same RW-based PE $\Phi_k$. In contrast, PH can separate them due to distinct connected components. Likewise, $K$ and $K'$ are 4-regular and 2-WL equivalent graphs which are indistinguishable due to the same RW-based PE.

In the following, we describe how we can leverage PE and PH methods to overcome their individual limitations and achieve enhanced representational capabilities.

### 3.1. Benefits of combining PE with PH

We now show that persistent homology benefits from additional expressive power due to positional encoding. We start by showing that defining filtering functions on positional encodings results in 0-dim persistence diagrams that are at least as expressive as the positional encodings in distinguishing non-isomorphic graphs. In other words, we do not lose expressive power by relying only on 0-dim diagrams. This is a direct consequence (corollary) of Lemma 5 by Immonen et al. (2023).

**Lemma 3.3.** *Let $\Phi(G) = \{p_v\}_{v \in V(G)}$ denote the node embeddings of $G$ from any base PE method. For any $G_1$ and $G_2$, if $\Phi(G_1) \neq \Phi(G_2)$, then there exists a function $f$ such that $\mathcal{D}^0(G_1, \Phi(G_1), f) \neq \mathcal{D}^0(G_2, \Phi(G_2), f)$.*

In Propositions 3.4 and 3.5, we show there are pairs of graphs that LapPE, random walk PE and distance encodings cannot distiguish but their combination with PH can. Importantly, together with Lemma 3.3, these results show that combining PH and PE is *strictly* more expressive than the base PE methods alone.

**Proposition 3.4.** *Let $\Phi_k(G)$ be either the Laplacian PEs with the $k$ lowest eigenpairs or the random walk PEs with walks of length $k$ of $G$. Then, there exist $G_1 \not\cong G_2$ and filtration function $f$ such that $\Phi_k(G_1) = \Phi_k(G_2)$ and $\mathcal{D}^0(G_1, \Phi_k(G_1), f) \neq \mathcal{D}^0(G_2, \Phi_k(G_2), f)$.*

To illustrate this limitation, consider graphs $G$ and $G'$ in Figure 4. Despite having different connected components, they share identical RW-based $\Phi_k$ for some $k$, demonstrating RW-based PE's inability to determine the number of connected components.

**Proposition 3.5.** *Let $\Phi_d(G)$ denote the distance encodings of $G$ considering the $d$-sized node subset $S \in \mathcal{P}_d(V(G))$. Then, there exist $G_1 \not\cong G_2$ and $f$ such that $\Phi_1(G_1) =$*

*$\Phi_1(G_2)$ and $\mathcal{D}^0(G_1, \Phi_1(G_1), f) \neq \mathcal{D}^0(G_2, \Phi_1(G_2), f)$.*

Additionally, Proposition 3.6 considers PH based on degree filtrations and shows that it does not subsume combinations of PH with LapPE and RWPE.

**Proposition 3.6.** *Consider $D(G) = \{d_v\}_{v \in V(G)}$ where $d_v$ is the degree of node $v$. Again, let $\Phi_k(G)$ be either the $k$-dim Laplacian PEs or $k$-length random walk PEs of $G$. Then, there exist $k > 0$, $G_1 \not\cong G_2$ such that $\mathcal{D}^0(G_1, D(G_1), f) = \mathcal{D}^0(G_2, D(G_2), f)$ for all $f$, but $\mathcal{D}^0(G_1, \Phi_k(G_1), f) \neq \mathcal{D}^0(G_2, \Phi_k(G_2), f)$ for some $f$.*

Figure 8 (in the Appendix) illustrates two graphs $G$ and $G'$ that are indistinguishable by degree-based filtration functions. Note that in contrast, the Laplacian eigenspectra are distinct, allowing LPE-based methods to distinguish them. Next, we present a *learnable* approach that combines PH, PE, and graph neural networks (GNNs), which are the most widely used methods for learning on graphs.

## 4. Persistence-informed positional encoding

In this section, we introduce *Persistence-informed positional encoding* (PiPE, in short). PiPE unifies PE with PH, via GNN-based message passing networks and leverages detailed topological information of graphs. Here, we also analyze the expressivity properties of our proposal.

Let $p_v \in \mathbb{R}^d$ be a base PE (e.g., Laplacian PE) for a node $v \in V(G)$. We propagate positional embeddings over the graph following a vanilla message-passing procedure while computing the topological features. In particular, starting from $p_v^0 = p_v$ for all $v$, we recursively update the positional embeddings as

$$r_v^{\ell,0} = \Psi_0^\ell(\mathcal{D}_\ell^0(A, \{p_v^\ell\}_v, f_\ell))_v \tag{3}$$

$$r_v^{\ell,1} = \sum_{e:v\in e} \Psi_1^\ell(\mathcal{D}_\ell^1(A, \{p_v^\ell\}_v, f_\ell))_e \tag{4}$$

$$p_v^{\ell+1} = \mathrm{Upd}_\ell^p(r_v^{\ell,0}, r_v^{\ell,1}, p_v^\ell, \\ \mathrm{Agg}_\ell(\{\!\{(r_u^{\ell,0}, r_u^{\ell,1}, p_u^\ell) : u \in \mathcal{N}(v)\}\!\})) \tag{5}$$

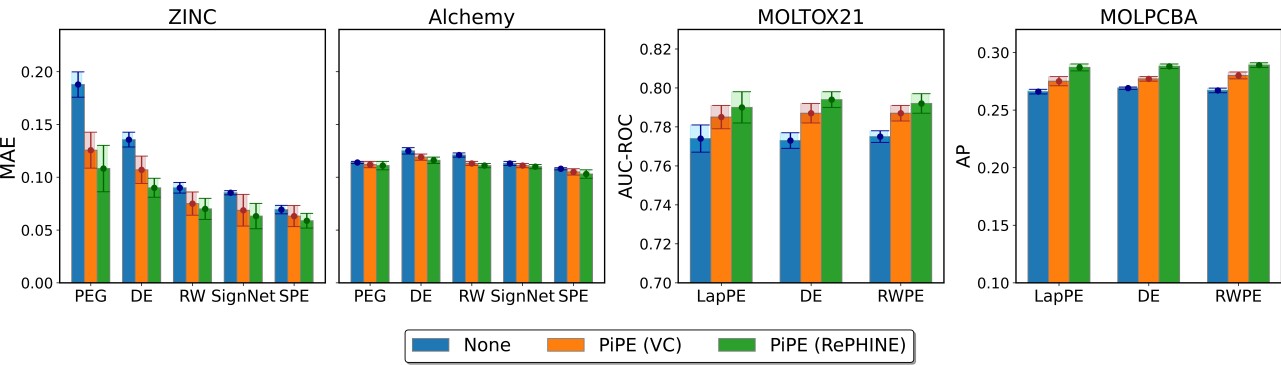

Figure 5: **Test MAE for drug molecule property prediction and graph classification results**. Integration of PiPE into backbone message passing schemes leads to better downstream performance across diverse datasets. For more details, see Table 5.

where $\mathrm{Agg}_\ell^p$ is an order-invariant function, $\mathrm{Upd}_\ell^p$ is an arbitrary update function at layer $\ell$, and $\Psi_0^\ell$ and $\Psi_1^\ell$ denote diagram vectorization schemes, detailed in the following.

**Computing topological features.** We use the positional encodings $\{p_v^\ell\}_v$ as node features to compute persistence diagrams $\mathcal{D}_\ell^0, \mathcal{D}_\ell^1$ induced by a filtering function $f_\ell$ followed by the vectorization procedures $\Psi_0^\ell, \Psi_1^\ell$ at each layer $\ell$. We followed the vectorization schemes in (Horn et al., 2022). To obtain node features from $\mathcal{D}_0$ we note that $|\mathcal{D}_0| = n$, and, therefore, we can define a bijection between $V$ and $\mathcal{D}_0$. Thus, we apply an MLP to each tuple to obtain the node-level features $r_v^{\ell,0}$. For dimension 1, we first employ the edge-level vectorization $\Psi_1^\ell$ (e.g., MLP) and then aggregate the edge embeddings to obtain node-level ones $r_v^{\ell,1}$. Note that, although $|\mathcal{D}_1|$ is equal to the number of basic cycles (not to the number of edges), we can use dummy tuples for edges that have not created a cycle. Details of this procedure can be found in Appendix A.4 in (Horn et al., 2022). We refer to $[r_v^{\ell,0} \,\|\, r_v^{\ell,1}]$ as the *topological embedding* associated with the base PE $p_v^\ell$.

**Integration with backbone GNNs.** A simple strategy to integrate PiPE with the backbone GNNs over node features is to combine (e.g., concatenate or add) them with GNN node embeddings $\{x_v^\ell\}_v$. Then, the resulting GNN's message-passing procedure at layer $\ell$ becomes

$$x_v^{\ell+1} = \mathrm{Upd}_\ell^x\left(h_v^\ell, \mathrm{Agg}_\ell(\{\!\!\{h_u^\ell : u \in \mathcal{N}(v)\}\!\!\})\right) \quad \forall v \in V.$$

where $h_v^\ell = [x_v^\ell \,\|\, p_v^\ell \,\|\, r_v^{\ell,0} \,\|\, r_v^{\ell,1}]$.

**Achieving class predictions.** For graph classification, as usual, we apply a readout function (e.g., sum or mean) to the embeddings at the last GNN layer, $L$, to obtain a graph-level embedding $x_G$, i.e., $x_G = \mathrm{Readout}(\{x_v^L\}_v)$. Similarly to LSPE, we can also concatenate positional embeddings $p_v^L$ with node representations $x_v^L$ before applying the readout function. Then, we combine $x_G$ with a global topological

embedding $\mathrm{Pool}(\{r_v^{\ell,0}, r_v^{\ell,1}\}_{\ell,v})$ and send the resulting vector through an MLP to achieve graph-level predictions — Pool is either a global mean or addition operator.

Figure 2 describes the architectural steps of our method. Importantly, our framework is versatile and can accommodate any selection of base (initial) positional encoding as well as various topological descriptors (e.g., RePHINE) and GNNs.

### 4.1. Analysis

We now report results on the expressiveness of LPE and RW-based PiPE. All proofs are in Appendix C.

In Propositions 4.1 and 4.2, we show that PiPE is strictly more expressive than simple PH and LSPE – a popular method for learnable positional encodings using GNNs.

**Proposition 4.1** (LPE-based PiPE $\succ$ LPE-based LSPE)**.** *Consider the space of unattributed graphs. Let $\mathcal{Q}$ and $\mathcal{J}$ be the classes of PiPE and LSPE models using Laplacian PE as base encoding, respectively. Then, $\mathcal{Q}$ is strictly more expressive than $\mathcal{J}$ in distinguishing non-isomorphic graphs.*

**Proposition 4.2** (LPE-based PiPE $\succ$ PH + LPE)**.** *Consider the space of unattributed graphs. Let $\mathcal{Q}$ be the class of PiPE models using Laplacian PE (LPE) as base encoding. Also, let $\mathcal{PH}$ be the class of models that computes persistence diagrams (dimensions 0 and 1) from filtrations induced by vertex colors derived from LPE. Then, $\mathcal{Q}$ is strictly more expressive than $\mathcal{PH}$ in distinguishing non-isomorphic graphs.*

Next, Proposition 4.3 describes a shortcoming of random walk-based PiPE: its inability to distinguish certain graph pairs that are separable by 3-WL.

**Proposition 4.3** (RW-based PiPE and 3-WL)**.** *There exists certain pair of non-isomorphic unattributed graphs, which can be distinguished by 3-WL but RW-based PiPE cannot.*

Figure 4 illustrates graphs $K$ and $K'$ that are 2-WL equiva-

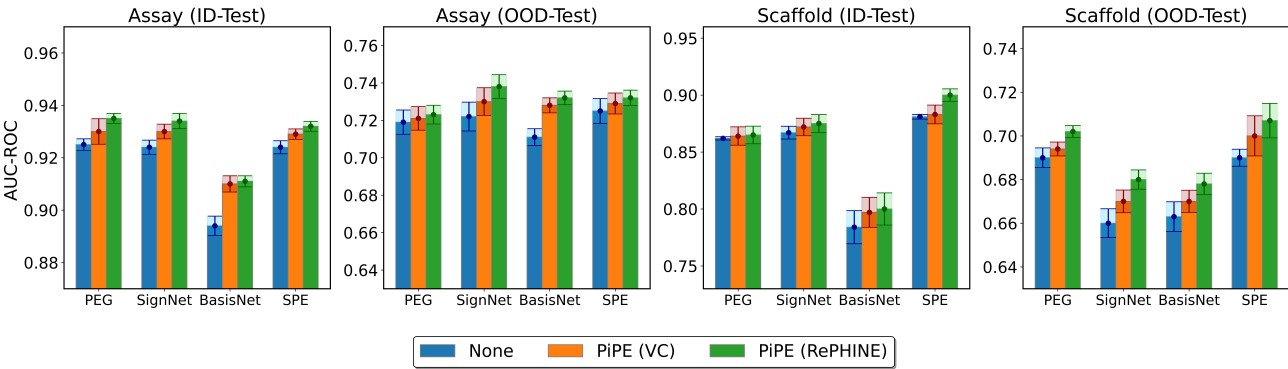

Figure 6: **AUC-ROC (↑) results for DrugOOD Benchmark**. **PiPE outperforms the competing baselines in achieving better scores for OOD-Test**. For more details, see Table 7.

lent, sharing identical node degrees and RW-based PE $\Phi_k$ for some $k$. Despite their structural differences, the PE method fail to distinguish between them.

Since PiPE combines GNNs with PH, we also provide a result on the connection between color-separating sets and the $k$-WL hierarchy. Proposition 4.4 shows that whenever $k$-FWL distinguishes two graphs, there exists a filtration that produces 0-dim persistence diagrams for these graphs, or equivalently, there is a color-separating set. We also provide an explicit coloring for the graphs based on the stable colorings from $k$-FWL.

**Proposition 4.4.** *If $k$-FWL deems two attributed graphs $(G, x)$ and $(G', x')$ non-isomorphic with stable colorings $C_\infty : V^k \to \mathbb{N}$ and $C'_\infty : V'^k \to \mathbb{N}$, then $Q = \emptyset$ is a trivial color-separating set for the graphs $(G, \tilde{x})$ and $(G', \tilde{x}')$, with $\tilde{x}(u) = \text{hash}(\{C_\infty(v) : u \in v, v \in V^k\}) \, \forall \, u \in V$ and $\tilde{x}'(u) = \text{hash}(\{C'_\infty(v) : u \in v, v \in V'^k\}) \, \forall \, u \in V'$.*

We note that Proposition 4.4 strengthens the results by Ballester & Rieck (2024) (Proposition 5) in two ways. First, we show how to use $k$-FWL to find a specific filtering function that gives separable diagrams — in fact, given the proposed coloring, separability holds for any injective vertex-color function. Also, with our scheme, even 0-dim diagrams are different, while Proposition 5 in (Ballester & Rieck, 2024) provides that $k - 1$ or $k$-dim diagrams are different.

## 5. Experiments

We assess the performance of PiPE on diverse and challenging tasks. In Section 5.1, we evaluate the expressivity of persistent homology and its combination with positional encoding on unattributed graphs. In Section 5.2, we assess its effectiveness in predicting properties of drug molecules and performing real-world graph classification. In Section 5.3 we evaluate PiPE's robustness by benchmarking its ability to handle domain shifts in data, and Section 5.4 shows the

performance of PiPE on synthetic tree-structured tasks.

**Implementation.** PiPE is implemented in PyTorch (Paszke et al., 2019) with same training configuration as the competing baselines. More details in Appendix D.

**Baselines.** To empirically demonstrate the effectiveness of our method, we compared it against existing positional encoding approaches on various tasks. We utilized several established baselines for graph tasks: (i) No positional encodings, (ii) SignNet & BasisNet (Lim et al., 2023), (iii) PEG (Wang et al., 2023), (iv) LapPE & RWPE (Dwivedi et al., 2022), (v) SPE (Huang et al., 2024) and (vi) DE (Li et al., 2020). In order to compute the topological descriptors via persistence homology, we utilized (i) Vertex Color (VC) (Horn et al., 2022) and (ii) RePHINE (Immonen et al., 2023) as learnable methods to compute the diagrams. For the synthetic tree tasks, we compared our method against these positional encoding approaches: (i) Sinusodial (Gehring et al., 2017), (ii) Relative (Shaw et al., 2018) and (iii) RoPE (Su et al., 2024) positional embedding methods.

Table 1: **Unattributed Graphs**. The table below reports accuracy, with the numbers in parentheses indicating the number of graph pairs in each dataset.

| Dataset | PH | PH+LPE | PiPE |
|---|---|---|---|
| Basic (60) | 0.03 | 0.10 | **0.72** |
| Regular (50) | 0.00 | 0.15 | **0.40** |
| Extension (100) | 0.07 | 0.13 | **0.67** |
| CFI (100) | 0.03 | 0.03 | 0.03 |
| Distance (20) | 0.00 | 0.00 | **0.05** |

### 5.1. Expressivity on Unattributed Graphs

We conducted an empirical study to assess the expressivity of standard 0-dim PH and PH+LPE on the BREC dataset (Wang & Zhang, 2023), which evaluates GNN expressive-

ness across four graph categories: Basic, Regular, Extension, and CFI. Table 1 presents the accuracy results for each graph category. The findings indicate that PH+LPE demonstrates greater expressiveness than standard PH, corroborating our theoretical analysis.

## 5.2. Drug Molecule Property Prediction and Graph Classification

We used the ZINC (Dwivedi et al., 2023) and Alchemy (Chen et al., 2019) datasets, containing quantum mechanical properties of drug molecules, with the aim to predict these properties. We followed the data preparation strategy of Huang et al. (2024) with GIN as the base model for a fair comparison. For graph classification, we used OGBG-MOLTOX21 (Huang et al., 2017; Wu et al., 2018), a multi-task binary classification dataset comprising of 7.8k molecular graphs for toxicity prediction across 12 measurements; OGBG-MOLHIV (Hu et al., 2020), which contains 41k molecular graphs with a binary classification task for HIV activity; and OGBG-MOLPCBA (Wang et al., 2012; Wu et al., 2018), a large-scale dataset with 437.9k graphs for predicting molecular activity or inactivity across 128 bioassays. We followed the experimental setup of Dwivedi et al. (2022), using Gated-GCN as the base model.

**Superior Results.** Figure 5 and Table 6 present the test Mean Absolute Error (MAE) for property prediction tasks (ZINC, Alchemy) and graph classification tasks (OGBG-MOLTOX21, OGBG-MOLPCBA) across various PH-vectorization schemes. Additionally, Table 6 also reports the ROC-AUC results on the OGBG-MOLHIV dataset. We observe that incorporating PiPE into PE schemes consistently outperforms baselines, particularly on ZINC and MOLPCBA, achieving notable improvements. Integrating PiPE with the PEG baseline yields the largest MAE reduction on ZINC, highlighting our approach's ability to capture richer graph representations.

## 5.3. Out of Distribution Prediction

To evaluate our method's ability to handle domain shifts, we used DrugOOD, an out-of-distribution (OOD) benchmark (Ji et al., 2023). We focused on the ligand-based affinity prediction task to assess drug activity. DrugOOD introduces two types of distribution shifts: (i) Assay, denoting the assay to which the data point belongs, and (ii) Scaffold, representing the core structure of molecules. The DrugOOD dataset is divided into five parts: training set, in-distribution (ID) validation/test sets, and out-of-distribution (OOD) validation/test sets. The OOD sets have different underlying distributions compared to the ID sets, allowing us to assess generalizability to unseen data.

**Superior OOD Generalizability.** Figure 6 summarizes the AUC-ROC scores for different methods over test datasets.

Interestingly, all models achieve similar performance on the in-distribution test set (ID-Test). However, performance drops for all methods on the out-of-distribution test set (OOD-Test). This highlights the challenge of generalizing to unseen data. Our method exhibits the best performance on the OOD-Test set. This demonstrates the effectiveness of our approach in capturing features relevant for generalizability, even when encountering unseen data distributions.

## 5.4. Synthetic Tree Tasks

We explored three synthetic tree-tasks involving binary branching trees: (i) Tree-copy, analogous to a sequence copy-task; (ii) Tree-rotation, where the output tree mirrors the input, interchanging left and right children; and (iii) Algebraic expression reduction, where input trees represent complex expressions from the cyclic group $C_3$, and the model is tasked with performing a single reduction step, i.e., reducing all depth-1 subtrees into leaves. We followed the data-preparation strategy of Kogkalidis et al. (2024) and utilized same splits and hyperparameters.

**Improved Performance on Tree Tasks.** Table 2 presents the Perplexity (PPL) scores for all methods on the synthetic tree tasks. Our method consistently achieves lower PPL scores compared to the baseline across all tasks. This indicates that incorporating our approach on top of a positional encoding method leads to improved performance on downstream tasks. This finding highlights the versatility of our method, demonstrating its effectiveness across various data domains, including those involving tree-structured data.

## 6. Ablations

**Identity Filtrations.** We investigated the impact of learnable versus non-learnable filtrations in vertex-color (VC) PH method. We compared using positional encodings directly via an identity filtration function (VC-I), to define filtration values for computing persistence diagrams, against a learned filtration function. Figure 7 shows the results alongside comparisons with learnable variants (VC & RePHINE) on ZINC dataset. We observe that using the positional encodings as filtration values to compute the persistence diagrams improves the performance. This is further enhanced by learning a parameterized filtration function, highlighting the method's increased expressiveness.

**Runtime Comparison.** We conducted an ablation study to investigate the computational cost of our method. We measured the time (in seconds) per epoch for different models on a single V100 GPU. The results for various PH and PE methods are shown in Figure 7 over the Alchemy dataset. We observe that SPE introduces additional computational overhead due to its more intensive computations compared to the simpler methods such as PEG. However, PiPE only

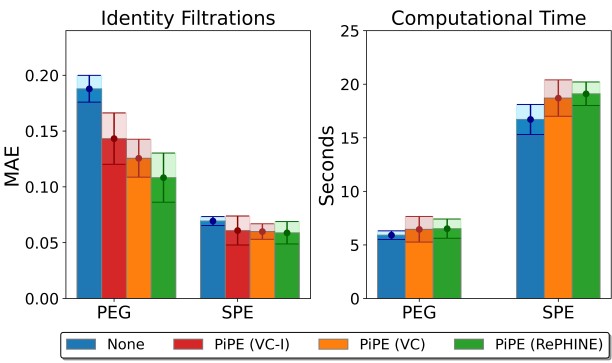

Figure 7: Identity filtrations and Runtime Comparisons.

Table 2: **Synthetic Tree tasks**. Perplexity (PPL) ↓ on synthetic tree tasks, where B stands for breadth and D for depth.

| Scheme | Persistence | C$_3$ | | Reorder | | Copy | |
|---|---|---|---|---|---|---|---|
| | | B | D | B | D | B | D |
| | None | 2.47 | 2.90 | 6.93 | **7.11** | 1.14 | 5.76 |
| Sinusodial | VC | 2.42 | 2.75 | 6.80 | 7.21 | 1.10 | 5.47 |
| | RePHINE | **2.33** | **2.64** | **6.75** | 7.51 | **1.00** | **5.32** |
| | None | 1.85 | 2.62 | 6.00 | **7.72** | 1.10 | 5.94 |
| Relative | VC | **1.53** | 2.42 | **5.92** | 8.11 | 1.01 | 5.04 |
| | RePHINE | 1.70 | **2.31** | 6.12 | 7.97 | **1.00** | **4.82** |
| | None | 1.84 | 2.52 | 4.93 | 6.63 | 1.85 | 3.17 |
| RoPE | VC | 1.65 | 1.94 | 4.76 | 5.24 | 1.14 | 2.35 |
| | RePHINE | **1.59** | **1.77** | **4.49** | **4.70** | **1.00** | **2.05** |

adds a slight overload over the base method.

## 7. Conclusion and Limitations

We highlight the incomparability of positional encoding and persistent homology methods. Building on these insights, we introduce "Persistence-informed Positional Encoding" (PiPE), a novel method that unifies the power of PH with general positional encoding methods. We theoretically analyze PiPE's expressive power and characterize its capabilities within the $k$-WL hierarchy. Our extensive empirical evaluations across diverse tasks demonstrate PiPE's effectiveness, showing significant improvements over existing methods. PiPE incurs computational overhead due to the cost of computing persistent homology (PH) embeddings. Moreover, combining message passing with persistence-augmented node representations may result in graph-level representations that are not permutation-invariant. Additionally, our current approach is limited to 1-dimensional simplicial complexes.

## Acknowledgements

This work has been supported by the Research Council of Finland under the *HEALED* project (grant 13342077), Jane and Aatos Erkko Foundation project (grant 7001703) on "Biodesign: Use of artificial intelligence in enzyme design for synthetic biology", and Finnish Center for Artificial Intelligence FCAI (Flagship programme). We acknowledge CSC – IT Center for Science, Finland, for providing generous computational resources. YV acknowledges the support from the Finnish Foundation for Technology Promotion (grant 10477).

## Impact Statement

Graph representation learning (GRL) is an area of active research, and finds use in important applications such as drug discovery. Our work advances the state of GRL, and thus opens exciting avenues. We are not aware of any specific negative societal consequences of this work.

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

## A. Related works

**Graph positional encodings.** Positional encodings enhance representations in Graph Neural Networks (GNNs) (Gilmer et al., 2017; Xu et al., 2019; Velickovic et al., 2018) by incorporating relational information between nodes based on their positions. Several approaches have been developed to achieve this, including Laplacian-based methods that utilize the graph laplacian (Dwivedi et al., 2022; Kreuzer et al., 2021; Maskey et al., 2022; Lim et al., 2023; Wang et al., 2023; Huang et al., 2024), random walk-based techniques that leverage walks on graphs (Dwivedi et al., 2022; Eliasof et al., 2023), and PageRank-inspired approaches that compute auxiliary distances (Ying et al., 2021; Li et al., 2020). However, these methods partition the Laplacian eigenvalue/eigenvector space and utilize only the partitioned eigenvalues/eigenvectors, disregarding the valuable information contained in the remaining eigenvalues and eigenvectors. To address this limitation, we propose complementing the existing approaches with topological descriptors based on persistent homology, which can capture additional structural information from the graph.

**Persistent homology on graphs** Persistence homology methods (Horn et al., 2022; Carriere et al., 2020; Immonen et al., 2023; Rosen et al., 2023; Hajij et al., 2021) from topological data analysis have made rapid strides, providing topological descriptors that augment GNNs (Cesa & Behboodi, 2023; Verma et al., 2024) with persistent information to obtain more powerful representations, enhancing the expressivity (Ballester & Rieck, 2024; Wang et al., 2024; Yan et al., 2025) and generalizability (Brilliantov et al., 2024). However, these methods have not been analyzed in regards with positional encodings in graphs, and the unification of these topological descriptors with positional encodings remains an unexplored frontier.

## B. WL Tests

The Weisfeiler–Leman test (1-WL), also known as the color refinement algorithm (Weisfeiler & Leman, 1968), aims to determine if two graphs are isomorphic. It does so by iteratively assigning colors to nodes. Initially, nodes receive labels based on their features. In each iteration, nodes sharing the same label get distinct labels if their sets of similarly labeled neighbors differ. Termination happens when label counts diverge between graphs, indicating non-isomorphism.

Due to the shortcomings of the 1-WL in distinguishing non-isomorphic graphs, Babai (1979); Immerman & Lander (1990) introduced a more powerful variant known as $k$-dim (*folklore*) Weisfeiler–Leman algorithm. In this approach, $k$-FWL colors subgraphs instead of a single node. Specifically, given a graph $G$, it colors tuples from $V(G)^k$ for $k \geq 1$ instead of nodes and defines neighborhoods between these tuples. Formally, let $G$ be a graph, and let $k \geq 2$. If $\mathbf{v} \in V(G)^k$, then $G[\mathbf{v}]$ is the subgraph induced by the components of $\mathbf{v}$, where the nodes are labeled with integers from $\{1, ..., k\}$ corresponding to indices of $\mathbf{v}$.

In each iteration $i \geq 0$, the algorithm computes a *coloring* $C_i^k : V(G)^k \to \mathbb{N}$, and in the initial iteration ($i = 0$) two tuples $\mathbf{v}$ and $\mathbf{w}$ in $V(G)^k$ get the same color if the map $v_i \to w_i$ induces an isomorphism between $G[\mathbf{v}]$ and $G[\mathbf{w}]$. For $i > 0$, the algorithm proceeds as,

$$C_{i+1}^k(\mathbf{v}) = \text{RELABEL}\left((C_i^k(\mathbf{v}), M(\mathbf{v}))\right) \tag{6}$$

where the multi-set $M(\mathbf{v}) = \{\!\!\{ C_i^k(\phi_1(\mathbf{v}, w)), \ldots, C_i^k(\phi_k(\mathbf{v}, w)) \mid w \in V(G) \}\!\!\}$ and $\phi_j(\mathbf{v}, w) = \{v_1, \ldots, v_{j-1}, w, v_{j+1}, v_k\}$. The $\phi_j(\mathbf{v}, w)$ replaces the $j$-th component of the tuple $\mathbf{v}$ with the node $w$. Consequently, two tuples are adjacent or $j$-neighbors (with respect to a node $w$) if they differ in the $j$th component (or are equal, in the case of self-loops). The algorithm iterates until convergence, i.e., $C_i^k(\mathbf{v}) = C_i^k(\mathbf{w}) \iff C_{i+1}^k(\mathbf{v}) = C_{i+1}^k(\mathbf{w})$ for all $\mathbf{v}$, defining the stable partition induced by $C_i^k$, define $C_\infty^k(\mathbf{v}) = C_i^k(\mathbf{v})$. The algorithm then proceeds analogously to the 1-WL.

We say that the $k$-FWL distinguishes two graphs $G$ and $H$ if their color histograms differ. This means there exist a color $c$ in the image of $C_\infty^k$ such that $G$ and $H$ have distinct numbers of node tuples of color $c$. Morris et al. (2023) also describe another variant of $k$-WL known as $k$-dim (*oblivious*) WL algorithm. The key distinction between the two lies in aggregating over different neighborhoods. In this case, for each position $j \in [k]$ we obtain a set of $|V(G)|$ neighbors by replacing $v_j$ by $w \in V$. A hash for position $j$ is obtained using these colors, and the overall color is obtained by aggregating the hashed colors across all $k$ positions (and $v$'s color from previous iteration). We utilize the former variant throughout the paper and refer to Morris et al. (2023); Huang & Villar (2021) for a thorough discussion of the algorithm and its properties.

# C. Proofs

## C.1. Proof of Proposition 3.1

**S1. There exist** $G_1 \not\cong G_2$ **s.t.** $\beta_0(G_1) = \beta_0(G_2)$, $\beta_1(G_1) = \beta_1(G_2)$ **but** $\Phi_k(G_1) \neq \Phi_k(G_2)$ **for all** $k > \beta_0(G_1)$

To prove this statement, we will provide a pair of graphs $G_1$ and $G_2$ with $\beta_0(G_1) = \beta_0(G_2)$ and $\beta_1(G_1) = \beta_1(G_2)$ for which $\Phi_k(G_1) \neq \Phi_k(G_2)$ with $k = \beta_0(G_1) + 1$. Naturally, if the graphs can be distinguished based on the $k'$ smallest eigenpairs, they are also distinguished based on any $k > k'$.

Consider the unattributed graph complexes $K = (V, E)$ and $K' = (V', E')$ as shown in Fig. 3, with associated positional encodings $\Phi_k(K)$ and $\Phi_k(K')$ based on $k = \beta_0(K) + 1$ lowest eigenvalue/vector pairs. Both $K$ and $K'$ has the same number of connected components i.e. $\beta_0(K) = \beta_0(K') = 1$, and basis cycles $\beta_1(K) = \beta_1(K') = 3$. However, the laplacian positional encoding based on $k = \beta_0(K) + 1$ lowest eigenvalues, for $K$ and $K'$ are

$$\Phi_k(K) = \begin{bmatrix} -0.50 & -0.32 \\ -0.50 & 0.32 \\ 0 & 0.53 \\ 0 & -0.53 \\ 0.50 & -0.32 \\ 0.50 & 0.32 \end{bmatrix}, \Phi_k(K') = \begin{bmatrix} -0.37 & 0 \\ -0.17 & 0.62 \\ -0.37 & 0 \\ -0.17 & -0.62 \\ 0.58 & -0.35 \\ 0.58 & 0.35 \end{bmatrix} \tag{7}$$

Hence, the positional encodings differ i.e. $\Phi_k(K) \neq \Phi_k(K')$.

**S2 & S3. There exist** $G_1 \not\cong G_2$ **s.t.** $\beta_1(G_1) \neq \beta_1(G_2)$, $\beta_0(G_1) \neq \beta_0(G_2)$ **but** $\Phi_k(G_1) = \Phi_k(G_2)$

Similarly to statement S1, here we proceed with a proof by example.

To prove this statement, we will provide a pair of graphs $G_1$ and $G_2$ with $\beta_0(G_1) \neq \beta_0(G_2)$ and $\beta_1(G_1) \neq \beta_1(G_2)$ for which $\Phi_k(G_1) = \Phi_k(G_2)$ for some $k$.

Let $K_i$ denote the complete graph with $i$ nodes. Consider a graph $G = \cup_{i=1}^{n/2} K_1 \cup K_3$ — here $K_1 \cup K_3$ denotes a graph with two components comprising one isolated node and a triangle i.e. $\beta_0(G) = n, \beta_1(G) = n/2$. Also, consider $G' = \cup_{i=1}^{n/2}(K_1 \cup K_1 \cup K_1 \cup K_1)$ — i.e., $4n/2$ isolated nodes i.e. $\beta_0(G') = 2n, \beta_1(G') = 0$, shown in Figure 8. The $k$ smallest eigenvalues corresponding to $G$ are all equal to 0 with the identical constant eigenvector, when $k \leq n$. Similarly, $G'$ has the same eigenvalues with identical constant eigenvectors. However, the number of connected components and basis cycles in $G$ and $G'$ differ, i.e. $\beta_1(G) \neq \beta_1(G')$ and $\beta_0(G) \neq \beta_0(G')$

## C.2. Proof of Proposition 3.2

**S1. There exist** $G_1 \not\cong G_2$ **s.t.** $\beta_0(G_1) = \beta_0(G_2)$, $\beta_1(G_1) = \beta_1(G_2)$ **but** $\Phi_k(G_1) \neq \Phi_k(G_2)$ **for all** $k > \beta_0(G_1)$

To prove this statement, we will provide a pair of graphs $G_1$ and $G_2$ with $\beta_0(G_1) = \beta_0(G_2)$ and $\beta_1(G_1) = \beta_1(G_2)$ for which $\Phi_k(G_1) \neq \Phi_k(G_2)$ for $k = \beta_0(G_1) + 1$. Naturally, if the graphs can be distinguished based on the $k' = \beta_0(G_1) + 1$, they are also distinguished based on any $k > k'$.

Consider the unattributed graph complexes $K = (V, E)$ and $K' = (V', E')$ as shown in Fig. 3, with associated positional encodings $\Phi_k(K)$ and $\Phi_k(K')$ based on $k = \beta_0(K) + 1$ length random walk positional encodings, where $\beta_0(K) = 1$. Both $K$ and $K'$ has the same number of connected components $\beta_0(K) = \beta_0(K') = 1$, and basis cycles $\beta_1(K) = \beta_1(K') = 3$. The positional encodings for both the graphs are,

$$\Phi_k(K) = \begin{bmatrix} 0 & 0.41 \\ 0 & 0.41 \\ 0 & 0.44 \\ 0 & 0.44 \\ 0 & 0.41 \\ 0 & 0.41 \end{bmatrix}, \Phi_k(K') = \begin{bmatrix} 0 & 0.33 \\ 0 & 0.50 \\ 0 & 0.33 \\ 0 & 0.50 \\ 0 & 0.41 \\ 0 & 0.41 \end{bmatrix} \tag{8}$$

Hence, the positional encodings differ i.e. $\Phi_k(K) \neq \Phi_k(K')$ for $k = \beta_0(K) + 1$.

**S2 & S3. There exist** $G_1 \not\cong G_2$ **s.t.** $\beta_1(G_1) \neq \beta_1(G_2)$, $\beta_0(G_1) \neq \beta_0(G_2)$ **but** $\Phi_k(G_1) = \Phi_k(G_2)$

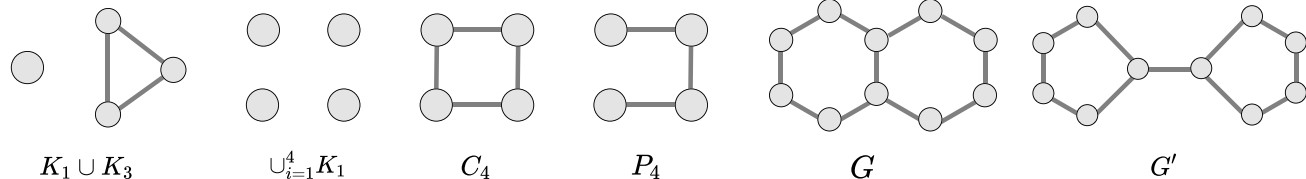

$$K_1 \cup K_3 \qquad \cup_{i=1}^{4} K_1 \qquad C_4 \qquad P_4 \qquad G \qquad G'$$

Figure 8: Exemplar graphs for Propositions 3.1, 3.4 and 4.2.

Similarly to statement S1, here we proceed with a proof by example.

To prove this statement, we will provide a pair of graphs $G_1$ and $G_2$ with $\beta_0(G_1) \neq \beta_0(G_2)$ and $\beta_1(G_1) \neq \beta_1(G_2)$ for which $\Phi_k(G_1) = \Phi_k(G_2)$ for some $k$.

Let $C_i$ denote the cyclic graph with $i$ nodes. Consider a graph $G = C_{10}$ having $\beta_0(G) = 1, \beta_1(G) = 1$, and $G' = C_5 \cup C_5$ (Figure 4) with $\beta_0(G') = 2, \beta_1(G') = 2$, both having a total 10 nodes, with associated positional encodings $\Phi_k(G)$ and $\Phi_k(G')$ based on $k = 4$ length random walk positional encodings. The positional encodings for both the graphs are

$$
\Phi_k(G) = \begin{bmatrix}
0 & 0.50 & 0 & 0.37 \\
0 & 0.50 & 0 & 0.37 \\
0 & 0.50 & 0 & 0.37 \\
0 & 0.50 & 0 & 0.37 \\
0 & 0.50 & 0 & 0.37 \\
0 & 0.50 & 0 & 0.37 \\
0 & 0.50 & 0 & 0.37 \\
0 & 0.50 & 0 & 0.37 \\
0 & 0.50 & 0 & 0.37 \\
0 & 0.50 & 0 & 0.37
\end{bmatrix}, \Phi_k(G') = \begin{bmatrix}
0 & 0.50 & 0 & 0.37 \\
0 & 0.50 & 0 & 0.37 \\
0 & 0.50 & 0 & 0.37 \\
0 & 0.50 & 0 & 0.37 \\
0 & 0.50 & 0 & 0.37 \\
0 & 0.50 & 0 & 0.37 \\
0 & 0.50 & 0 & 0.37 \\
0 & 0.50 & 0 & 0.37 \\
0 & 0.50 & 0 & 0.37 \\
0 & 0.50 & 0 & 0.37
\end{bmatrix}
\tag{9}
$$

Hence, the positional encodings for both the graphs are same i.e., $\Phi_k(G) = \Phi_k(G')$, but the number of connected components and basis cycle differ i.e. $\beta_0(G) \neq \beta_0(G')$ and $\beta_1(G) \neq \beta_1(G')$.

### C.3. Proof of Lemma 3.3

To prove Lemma 3.3, we need to show that the persistence diagram pairs obtained when using $\Phi(G)$ and $\Phi(G')$ as vertex colors are different. Importantly, Lemma 5 in Immonen et al. (2023) states that the multiset of birth times of persistence tuples encodes the multiset of vertex colors in VC filtrations when using injective filtration functions.

Therefore, if we use $\Phi(G_1)$ and $\Phi(G_2)$ as colors and adopt an injective filtration function, then the corresponding VC diagrams $\mathcal{D}^0(G_1, \Phi(G_1), f)$ and $\mathcal{D}^0(G_2, \Phi(G_2), f)$ differ in their birth times since $\Phi(G_1) \neq \Phi(G_2)$. This is agnostic to the chosen graphs and positional encoding.

### C.4. Proof of Proposition 3.4

### (1) Laplacian PE

To prove this statement, we will provide a pair of graphs $G_1$ and $G_2$ with for which $\Phi_k(G_1) = \Phi_k(G_2)$ for some $k$, but the persistence diagrams of $G_1$ and $G_2$ differ.

Let $K_i$ denote the complete graph with $i$ nodes. Consider a graph $G = \cup_{i=1}^{n/2} K_1 \cup K_3$ — here $K_1 \cup K_3$ denotes a graph with two components comprising one isolated node and a triangle. Also, consider $G' = \cup_{i=1}^{n/2}(K_1 \cup K_1 \cup K_1 \cup K_1)$ — i.e., $4n/2$ isolated nodes, shown in Figure 8. The $k$ smallest eigenvalues corresponding to $G$ are all equal to 0 with the identical constant eigenvector, for $k \leq n$. Similarly, $G'$ has the same eigenvalues with identical constant eigenvectors. Therefore, Laplacian PE relying on fixed $k$ smallest eigenvalue/eigenvector pairs, cannot distinguish these graphs.

By leveraging Theorem 1 from Immonen et al. (2023), since the number of connected components in $G$ and $G'$ are different, they necessarily have different 0-dimensional persistence diagrams i.e., $\mathcal{D}^0(G, \Phi_k(G), f) \neq \mathcal{D}^0(G', \Phi_k(G'), f)$ for any injective color-filtration function $f$ over vertices and using $\Phi_k(G)$ and $\Phi_k(G')$ as colors. This difference in persistence

diagrams allows us to distinguish between the graphs despite their identical $n$ smallest eigenvalues and eigenvectors.

**(2) Random Walk PE**

Similarly to the above statement, here we proceed with a proof by example.

Consider a graph $G = C_{10}$ having $\beta_0(G) = 1, \beta_1(G) = 1$, and $G' = C_5 \cup C_5$ (Figure 4) with $\beta_0(G') = 2, \beta_1(G') = 2$, both having a total 10 nodes, with associated positional encodings $\Phi_k(G)$ and $\Phi_k(G')$ based on $k = 4$ length random walk positional encodings. The positional encodings for both the graphs are same i.e., $\Phi_k(G) = \Phi_k(G')$ as shown in Equation (9). However, the number of connected components ($\beta_0$) in $G$ and $G'$ differ. Hence, by leveraging Theorem 1 from Immonen et al. (2023) and using the $\Phi_k$ as initial colors will provide different 0-dimensional persistence diagrams i.e., $\mathcal{D}^0(G, \Phi_k(G'), f) \neq \mathcal{D}^0(G', \Phi_k(G'), f)$ for any injective color-filtration function $f$ over vertices. Hence, PH $\circ$ RW can separate these graphs.

### C.5. Proof of Proposition 3.6

**(1) Laplacian PE**

To prove this statement, we will provide a pair of graphs $G_1$ and $G_2$ for which $\mathcal{D}^0(G_1, D(G_1), f) = \mathcal{D}^0(G_2, D(G_2), f)$ for all $f$ but have different diagrams $\mathcal{D}^0(G_1, \Phi_k(G_1), f) = \mathcal{D}^0(G_2, \Phi_k(G_2), f)$ derived from LapPE colors $\Phi_k(G_1)$ and $\Phi_k(G_1)$ for some $k$ and $f$.

Consider the unattributed graph complexes $G = (V, E)$ and $G' = (V', E')$ as shown in Fig. 8, with associated positional encodings $\Phi_k(G)$ and $\Phi_k(G')$ based on $k$ lowest eigenvalue/vector pairs, for $k = \beta_0(G) + 1$, where $\beta_0(G) = 1$. Using degree as the initial colors and adopting an injective filtration function $f$, we can have the following two scenarios: (i) $\gamma > \alpha$, and (ii) $\gamma < \alpha$, where $\gamma = f(d = 2)$ and $\alpha = f(d = 3)$ — where $d$ denotes node degree. In both cases, the obtained diagrams for both $G$ and $G'$ are identical, i.e.,

$$\mathcal{D}^0(G, D(G), f) = \begin{cases} \{(\alpha, \alpha), (\alpha, \infty), 8 \times (\gamma, \gamma)\} & \text{if } \gamma > \alpha \\ \{8 \times (\alpha, \alpha), (\gamma, \infty), (\gamma, \gamma)\} & \text{if } \gamma < \alpha \end{cases} \tag{10}$$

$$\mathcal{D}^0(G', D(G'), f) = \begin{cases} \{(\alpha, \alpha), (\alpha, \infty), 8 \times (\gamma, \gamma)\} & \text{if } \gamma > \alpha \\ \{8 \times (\alpha, \alpha), (\gamma, \infty), (\gamma, \gamma)\} & \text{if } \gamma < \alpha \end{cases} \tag{11}$$

Thus, $\mathcal{D}^0(G, D(G), f) = \mathcal{D}^0(G', D(G'), f)$ — PH relying on degree information cannot distinguish these graphs.

However, the Laplacian positional encodings for these graphs are different:

$$\Phi_k(G) = \begin{bmatrix} -0.42 & -0.18 \\ -0.42 & 0.18 \\ -0.26 & 0.39 \\ 0.00 & 0.34 \\ 0.26 & 0.39 \\ 0.42 & 0.18 \\ 0.42 & -0.18 \\ 0.26 & -0.39 \\ -0.00 & -0.34 \\ -0.26 & -0.39 \end{bmatrix}, \Phi_k(G') = \begin{bmatrix} -0.37 & 0.35 \\ -0.37 & 0.35 \\ -0.29 & -0.05 \\ -0.19 & -0.49 \\ -0.29 & -0.05 \\ 0.19 & -0.49 \\ 0.29 & -0.05 \\ 0.37 & 0.35 \\ 0.37 & 0.35 \\ 0.29 & -0.05 \end{bmatrix} \tag{12}$$

By leveraging Theorem 1 from Immonen et al. (2023) and Lemma 3.3, using the $\Phi_k$ as colors and adopting an injective filtration function, then the corresponding VC diagrams $\mathcal{D}^0(G, \Phi_k(G), f)$ and $\mathcal{D}^0(G', \Phi_k(G'), f)$ differ in their birthtimes. This difference will help to distinguish between these graphs.

**(2) Random Walk PE**

Similarly to the above statement, here we proceed with a proof by example.

We follow the same proof for $G$ and $G'$ having the same $0-$dim diagrams based on degree-based filtration functions, as described in the previous statement.

The associated RW positional encodings for both graphs differ i.e. $\Phi_k(G) \neq \Phi_k(G')$ for $k = 5$ length, as shown in Equation (13).

$$
\Phi_k(G) = \begin{bmatrix} 0 & 0.50 & 0 & 0.35 & 0 \\ 0 & 0.50 & 0 & 0.35 & 0 \\ 0 & 0.41 & 0 & 0.28 & 0 \\ 0 & 0.44 & 0 & 0.31 & 0 \\ 0 & 0.41 & 0 & 0.28 & 0 \\ 0 & 0.50 & 0 & 0.35 & 0 \\ 0 & 0.50 & 0 & 0.35 & 0 \\ 0 & 0.41 & 0 & 0.28 & 0 \\ 0 & 0.44 & 0 & 0.31 & 0 \\ 0 & 0.41 & 0 & 0.28 & 0 \end{bmatrix}, \Phi_k(G') = \begin{bmatrix} 0 & 0.50 & 0 & 0.35 & 0.04 \\ 0 & 0.50 & 0 & 0.35 & 0.04 \\ 0 & 0.41 & 0 & 0.28 & 0.04 \\ 0 & 0.44 & 0 & 0.31 & 0.04 \\ 0 & 0.41 & 0 & 0.28 & 0.04 \\ 0 & 0.44 & 0 & 0.31 & 0.04 \\ 0 & 0.41 & 0 & 0.28 & 0.04 \\ 0 & 0.50 & 0 & 0.35 & 0.04 \\ 0 & 0.50 & 0 & 0.35 & 0.04 \\ 0 & 0.41 & 0 & 0.28 & 0.04 \end{bmatrix} \tag{13}
$$

Again, by leveraging Theorem 1 from Immonen et al. (2023) and Lemma 3.3, using injective filtration functions on these colors leads to diagrams that differ in their birth times.

### C.6. Proof of Proposition 3.5

To prove this statement, we will provide a pair of graphs $G_1$ and $G_2$ with for which $\Phi_d(G_1) = \Phi_d(G_2)$ for $d = 1$, but the persistence diagrams of $G_1$ and $G_2$ differ.

Let $Q_i$ denote a hypercube graph with $i$ nodes. Consider a graph $G = Q_3$ comprising of $2^3$ nodes, with $\beta_0(G) = 1$. Also, consider $G' = Q_2 \cup Q_2$, with $\beta_0(G') = 2$ — consisting of two $Q_2$ graphs, having a total of $2^2 + 2^2$ nodes . Since, $Q_2$ and $Q_3$ are distance regular graphs (Brouwer et al., 2012), both $G$ and $G'$ have the same DE-1 i.e. $\Phi_d(G) = \Phi_d(G')$ for each node in the graph with $d = 1$. Therefore, distance encoding relying on shortest path distance, cannot distinguish these graphs.

By leveraging Theorem 1 from Immonen et al. (2023), since the number of connected components in $G$ and $G'$ are different i.e. $\beta_0(G) \neq \beta_0(G')$, they necessarily have different 0-dimensional persistence diagrams i.e., $\mathcal{D}^0(G, \Phi_d(G), f) \neq \mathcal{D}^0(G', \Phi_d(G'), f)$ for any color-filtration function over vertices and initial colors.

### C.7. Proof of Proposition 4.1

To prove this statement, it suffices to show that i) PiPE subsumes LSPE for certain choices of Agg, Upd functions , and (ii) there exists a pair of graphs which cannot be separated by LSPE but PiPE can. Note that LSPE can be seen as GNN $\circ$ LPE, and we consider unattributed graphs here.

**PiPE subsumes LSPE**  The message passing steps of PiPE are shown below, where $h_v^\ell = [x_v^\ell \parallel p_v^\ell \parallel r_v^{\ell,0} \parallel r_v^{\ell,1}]$.

$$z_v = f_\ell(p_v^\ell) \tag{14}$$

$$r_v^{\ell,0} = \Psi_0^\ell(\mathcal{D}_\ell^0(A, z))_v \tag{15}$$

$$r_v^{\ell,1} = \sum_{e:v \in e} \Psi_1^\ell(\mathcal{D}_\ell^1(A, z))_e \tag{16}$$

$$p_v^{\ell+1} = \mathrm{Upd}_\ell^p(r_v^{\ell,0}, r_v^{\ell,1}, p_v^\ell, \mathrm{Agg}_\ell^p(\{\!\!\{(r_u^{\ell,0}, r_u^{\ell,1}, p_u^\ell) : u \in \mathcal{N}(v)\}\!\!\})) \quad \forall\, v \in V \tag{17}$$

$$x_v^{\ell+1} = \mathrm{Upd}_\ell^x\left(h_v^\ell, \mathrm{Agg}_\ell^x(\{\!\!\{h_u^\ell : u \in \mathcal{N}(v)\}\!\!\})\right) \quad \forall\, v \in V. \tag{18}$$

Consider the following choices for the $\mathrm{Agg}_\ell^{x,p}$ and $\mathrm{Upd}_\ell^{x,p}$,

$$
\mathrm{PiPE} = \begin{cases} \mathrm{Agg}_\ell^x = \mathrm{Agg}_\ell^{x,\mathrm{LSPE}} \circ m_\ell^x, \\ \mathrm{Agg}_\ell^p = \mathrm{Agg}_\ell^{p,\mathrm{LSPE}} \circ m_\ell^p, \\ \mathrm{Upd}_\ell^p = \mathrm{Upd}_\ell^{p,\mathrm{LSPE}} \circ m_\ell'^p, \\ \mathrm{Upd}_\ell^x = \mathrm{Upd}_\ell^{x,\mathrm{LSPE}} \circ m_\ell'^x \end{cases}
$$

where $\mathrm{Agg}_\ell^{x,p,\mathrm{LSPE}}$ and $\mathrm{Upd}_\ell^{x,p,\mathrm{LSPE}}$ are the aggregate and update functions used in LSPE, and $m_\ell^{x,p}, m_\ell'^{x,p}$ are the masking function that that masks out the topological embeddings $r_v^{\ell,0}, r_v^{\ell,1}$ at every layer of the message-passing step. These choices for the functions will lead to the same message-passing dynamics of LSPE neglecting the topological features. Hence, this shows that PiPE subsumes LSPE.

**PiPE > LSPE**    Let $C_i$ denote a cycle graph with $i$ nodes without node attributes. Consider a graph $G = \cup_{i=1}^{3n} C_4$, having $3n$ connected components and $12n$ nodes. Also, consider $G' = \cup_{i=1}^n C_6 \cup C_6$, having $12n$ nodes and $2n$ connected components. Since, $G$ and $G'$ have the same number of nodes and identical local neighborhoods, aggregate-combine GNN cannot distinguish them. Moreover, for $k \leq 12n$, the $\Phi_k(G) = \Phi_k(G')$, are same having identical constant eigenvector,thus LPE does not add any distinguishability power.

However, since the number of components in $G$ and $G'$ are different, they have different 0-dimensional persistence diagram $\mathcal{D}^0(G, \Phi_k(G), f) \neq \mathcal{D}^0(G', \Phi_k(G'), f)$ for any filtration function and initial colors. This difference in persistence diagrams allows PiPE to distinguish between the graphs.

## C.8. Proof of Proposition 4.2

To prove this, it suffices to show that i) PiPE subsumes PH + LPE , and (ii) there exists a pair of graphs which cannot be separated by PH + LPE but PiPE can.

**PiPE subsumes PH+LPE**    We can write PiPE as GNN ∘ PH ∘ LPE. Consider, the GNN to be an identity map, then it will reduce to PH ∘ LPE. Hence, PiPE subsumes PH+LPE.

**PiPE > PH+LPE**    Let $C_i$ denote a cycle graph with $i$ nodes without node attributes and $P_i$ denotes a path graph with $i$ nodes without node attributes. Consider a graph $G = \cup_{i=1}^n C_4$ and $G' = \cup_{i=1}^n P_4$ (shown in Figure 8), which has $n$ connected components and $4n$ nodes. The $k$ smallest laplcian eigenvalues corresponding to $G$ and $G'$ are all equal to 0 with the identical constant eigenvector, where $k < 4n$, and both of the graphs consists of same number of connected components. Hence, by leveraging Theorem 1 from Immonen et al. (2023) the filtrations based on laplaician eigen value and vector pairs would correspond to the same 0-dimensional diagram, i.e. $\mathcal{D}^0(G, \Phi_k(G), f) = \mathcal{D}^0(G', \Phi_k(G'), f)$ for any filtration function and initial colors. Hence, Laplacian eigen spectra relying on partial-decomposition and relying on 0-dim persistence diagrams would not be able to separate these two graphs.

However, the nodes in $G$ and $G'$ have different degrees, allowing GNNs (in PiPE) to distinguish these two graphs.

## C.9. Proof of Proposition 4.3

To prove this, it suffices to show that a pair of graphs that can be separated by 3-WL have the same random walk positional encoding $\Phi_k$ for $k > 0$.

Consider the graph representation of pair of cospectral and 4-regular graphs $K$ and $K'$ (Van Dam & Haemers, 2003) from Figure 4, with the associated positional encodings $\Phi_k$ obtained via random walk PE for $k = 4$. These pair of graphs are 2-WL equivalent (Balcilar et al., 2021) i.e., these two graphs cannot be separated by 2-WL but 3-WL can separate them. The positional encodings for both the graphs are

$$\Phi_k(K) = \begin{bmatrix} 0 & 0.25 & 0.62 & 0.14 \\ 0 & 0.25 & 0.62 & 0.14 \\ 0 & 0.25 & 0.62 & 0.14 \\ 0 & 0.25 & 0.93 & 0.14 \\ 0 & 0.25 & 0.93 & 0.14 \\ 0 & 0.25 & 0.62 & 0.14 \\ 0 & 0.25 & 0.62 & 0.14 \\ 0 & 0.25 & 0.93 & 0.14 \\ 0 & 0.25 & 0.62 & 0.14 \\ 0 & 0.25 & 0.93 & 0.14 \end{bmatrix}, \Phi_k(K') = \begin{bmatrix} 0 & 0.25 & 0.93 & 0.14 \\ 0 & 0.25 & 0.62 & 0.14 \\ 0 & 0.25 & 0.93 & 0.14 \\ 0 & 0.25 & 0.93 & 0.14 \\ 0 & 0.25 & 0.93 & 0.14 \\ 0 & 0.25 & 0.62 & 0.14 \\ 0 & 0.25 & 0.62 & 0.14 \\ 0 & 0.25 & 0.62 & 0.14 \\ 0 & 0.25 & 0.62 & 0.14 \\ 0 & 0.25 & 0.62 & 0.14 \end{bmatrix} \tag{19}$$

Hence, the positional encodings for both the graphs are same i.e., $\Phi_k(K) = \Phi_k(K')$ with $\beta_0(K) = \beta_0(K') = 1$.Thus, using positional encodings $\Phi_k$ as initial colors with an injectve filtration function will lead to the same VC persistence

diagrams, that is, $\mathcal{D}^0(K, \Phi_k(K), f) = \mathcal{D}^0(K', \Phi_k(K'), f)$. Thus, PiPE based on random walk positional encodings cannot separate these graphs which can be separated by 3-WL.

### C.10. Proof of Proposition 4.4

Consider two attributed graphs $G = (V, E)$ and $G' = (V', E')$ with their corresponding coloring functions $x$ and $x'$. Assume these graphs are deemed non-isomorphic by $k$-FWL with stable colorings, $C_\infty : V^k \to \mathbb{N}$ and $C'_\infty : V'^k \to \mathbb{N}$.

Then we can use hash functions $\tilde{x}(u) = \text{hash}(\{C_\infty(v) : u \in v, v \in V^k\})$ for all $u \in V$ and $\tilde{x}'(u) = \text{hash}(\{C'_\infty(v) : u \in v, v \in V'^k\})$ for all $u \in V'$, to project the colorings from $k$-tuple of nodes to obtain node colors in the associated graphs $(G, \tilde{x})$ and $(G', \tilde{x}')$. Since, hash functions are injective in nature, and $(G, x)$ and $(G', x')$ can be distinguished via $k$-FWL, then there exists a tuple $\mathbf{v}_w \in V^k$ such that $C_\infty(\mathbf{v}_w) \neq C'_\infty(\mathbf{v}'_w)$, $\forall \mathbf{v}'_w \in V'^k$. Then, any node in $v \in \mathbf{v}_w$ (note that $v \in V$) will have a color that is not in nodes of $V'$, i.e., $\tilde{x}(v) \neq \tilde{x}'(u)$ for all $u \in V'$. This will provide us with different node colors for the associated graphs $(G, \tilde{x})$ and $(G', \tilde{x}')$. Hence, by leveraging the definition of color-separating sets from Immonen et al. (2023), $Q = \emptyset$ is a trivial color-separating set for these graphs.

## D. Implementation Details

Below are the implementation details. We trained all the methods on a single NVIDIA V100 GPU.

### D.1. Drug Molecule Property prediction and Graph Classification

We adhered to the precise hyperparameters and training configuration outlined in Huang et al. (2024) for predicting drug molecule properties and in Dwivedi et al. (2022) for classifying real-world graphs in our experiments. For graph classification, we used Gated-GCN (Kipf & Welling, 2017) as our base model. To compute the Persistence Homology (PH) diagrams, we employed the learnable PH method proposed by Immonen et al. (2023). The PH layers operated exclusively on the position encoding features of every layer with the following specified hyperparameters in Table 3.

Table 3: Default hyperparameters for RePHINE/VC method

| Hyperparameter | Meaning | Value |
|---|---|---|
| PH embed dim | Latent dimension of PH features | 64 |
| Num Filt | Number of filtrations | 8 |
| Hiden Filtration | Hidden dimension of filtration functions | 16 |

### D.2. Out of distribution Prediction

We adhered to the precise hyperparameters and training configuration outlined in Huang et al. (2024) for Drug OOD benchmark. To compute the Persistence Homology (PH) diagrams, we employed the learnable PH method proposed by Immonen et al. (2023). The PH layers operated exclusively on the position encoding features of every layer with the following specified hyperparameters in Table 3.

### D.3. Synthetic Tree Tasks

We created the synthetic tree dataset by sampling random trees of maximum depths from a discretized normal $\mathcal{N}(7, 1)$ and followed similar training setup as described in Kogkalidis et al. (2024). We adhered to the hyper-parameters and training configuration used in Kogkalidis et al. (2024) and employed the PH-layers on top of the position encoding features with an additional layer to update position encodings, using hyper-parameters described in Table 4.

## E. Tabular Results

Table 4: Default hyperparameters for RePHINE/VC method

| Hyperparameter | Meaning | Value |
|---|---|---|
| PH embed dim | Latent dimension of PH features | 64 |
| Num Filt | Number of filtrations | 8 |
| Hiden Filtration | Hidden dimension of filtration functions | 128 |

Table 5: **Test MAE results**. Baselines are taken from Huang et al. (2024).

| PE method | Persistence | ZINC ↓ | Alchemy ↓ |
|---|---|---|---|
| None | None | 0.1772 ±0.004 | 0.112 ±0.001 |
| PEG | None | 0.1878 ±0.012 | 0.114 ±0.001 |
| PiPE | VC | 0.1256 ±0.017 | 0.112 ±0.003 |
| | RePHINE | **0.1082** ±0.022 | **0.111** ±0.004 |
| DE | None | 0.1356 ±0.007 | 0.125 ±0.003 |
| PiPE | VC | 0.107 ±0.013 | 0.119 ±0.003 |
| | RePHINE | **0.090** ±0.009 | **0.116** ±0.003 |
| RW | None | 0.090 ±0.005 | 0.121 ±0.002 |
| PiPE | VC | 0.075 ±0.011 | 0.113 ±0.002 |
| | RePHINE | **0.070** ±0.010 | **0.111** ±0.002 |
| SignNet | None | 0.0853 ±0.002 | 0.113 ±0.002 |
| PiPE | VC | 0.0687 ±0.015 | 0.111 ±0.002 |
| | RePHINE | **0.0632** ±0.012 | **0.110** ±0.002 |
| SPE | None | 0.0693 ±0.004 | 0.108 ±0.001 |
| PiPE | VC | 0.0599 ±0.010 | 0.105 ±0.003 |
| | RePHINE | **0.0588** ±0.007 | **0.103** ±0.004 |

Table 6: (*left*) AUC-ROC results on OGBG-MOLHIV and (*right*) Test MAE results for TOGL.

| PE method | Persistence | OGBG-MOLHIV ↑ |
|---|---|---|
| RW | None | 0.762 ±0.007 |
| PiPE | VC | 0.781 ±0.005 |
| | RePHINE | **0.798** ±0.004 |
| SPE | None | 0.776 ±0.004 |
| PiPE | VC | 0.785 ±0.005 |
| | RePHINE | **0.791** ±0.003 |

| PE method | Persistence | ZINC ↓ | Alchemy ↓ |
|---|---|---|---|
| RW | TOGL | 0.080 ±0.005 | 0.114 ±0.002 |
| PEG | TOGL | 0.143 ±0.012 | 0.112 ±0.002 |
| SPE | TOGL | 0.062 ±0.003 | 0.112 ±0.002 |

Table 7: **AUC-ROC (↑) results**. DrugOOD Benchmark and baselines are taken from Huang et al. (2024). **PiPE outperforms the competing baselines in achieving better scores for OOD-Test.**

| Domain | PE method | Persistence | ID-Val ↑ | ID-Test↑ | OOD-Val ↑ | OOD-Test↑ |
|---|---|---|---|---|---|---|
| Assay | None | None | $92.92_{\pm0.14}$ | $92.89_{\pm0.14}$ | $71.02_{\pm0.79}$ | $71.68_{\pm1.10}$ |
| | PEG | None | $92.51_{\pm0.17}$ | $92.57_{\pm0.22}$ | $70.86_{\pm0.44}$ | $71.98_{\pm0.65}$ |
| | PiPE | VC | $92.62_{\pm0.19}$ | $92.75_{\pm0.49}$ | $71.62_{\pm0.57}$ | $72.13_{\pm0.93}$ |
| | | RePHINE | $92.42_{\pm0.27}$ | $92.35_{\pm0.19}$ | $72.02_{\pm0.51}$ | $\mathbf{72.33}_{\pm1.03}$ |
| | SignNet | None | $92.26_{\pm0.21}$ | $92.43_{\pm0.27}$ | $70.16_{\pm0.56}$ | $72.27_{\pm0.97}$ |
| | PiPE | VC | $91.66_{\pm0.39}$ | $92.73_{\pm0.28}$ | $70.37_{\pm0.69}$ | $73.07_{\pm1.07}$ |
| | | RePHINE | $91.36_{\pm0.31}$ | $92.15_{\pm0.29}$ | $69.47_{\pm0.43}$ | $\mathbf{73.87}_{\pm1.32}$ |
| | BasisNet | None | $88.96_{\pm1.35}$ | $89.42_{\pm1.18}$ | $71.19_{\pm0.72}$ | $71.16_{\pm0.05}$ |
| | PiPE | VC | $90.36_{\pm0.65}$ | $90.78_{\pm1.21}$ | $72.76_{\pm1.32}$ | $72.98_{\pm0.10}$ |
| | | RePHINE | $90.73_{\pm0.45}$ | $91.10_{\pm0.92}$ | $72.98_{\pm0.65}$ | $\mathbf{73.10}_{\pm0.25}$ |
| | SPE | None | $92.84_{\pm0.20}$ | $92.94_{\pm0.15}$ | $71.26_{\pm0.62}$ | $72.53_{\pm0.66}$ |
| | PiPE | VC | $92.78_{\pm0.96}$ | $92.49_{\pm0.58}$ | $71.78_{\pm0.64}$ | $72.91_{\pm1.16}$ |
| | | RePHINE | $92.16_{\pm0.37}$ | $93.12_{\pm0.91}$ | $72.33_{\pm0.93}$ | $\mathbf{73.11}_{\pm1.07}$ |
| Scaffold | None | None | $96.56_{\pm0.10}$ | $87.95_{\pm0.20}$ | $79.07_{\pm0.97}$ | $68.00_{\pm0.60}$ |
| | PEG | None | $95.65_{\pm0.29}$ | $86.20_{\pm0.14}$ | $79.17_{\pm0.29}$ | $69.15_{\pm0.75}$ |
| | PiPE | VC | $96.65_{\pm0.31}$ | $86.44_{\pm0.81}$ | $79.79_{\pm0.47}$ | $\mathbf{70.12}_{\pm0.52}$ |
| | | RePHINE | $96.94_{\pm0.62}$ | $86.54_{\pm0.77}$ | $79.40_{\pm0.35}$ | $69.31_{\pm0.97}$ |
| | SignNet | None | $95.48_{\pm0.34}$ | $86.73_{\pm0.56}$ | $77.81_{\pm0.70}$ | $66.43_{\pm1.06}$ |
| | PiPE | VC | $93.03_{\pm0.57}$ | $83.65_{\pm0.77}$ | $74.73_{\pm0.65}$ | $67.37_{\pm1.12}$ |
| | | RePHINE | $93.35_{\pm0.56}$ | $85.05_{\pm0.79}$ | $75.05_{\pm1.04}$ | $\mathbf{68.03}_{\pm1.34}$ |
| | BasisNet | None | $85.80_{\pm3.75}$ | $78.44_{\pm2.45}$ | $73.36_{\pm1.44}$ | $66.32_{\pm5.68}$ |
| | PiPE | VC | $86.10_{\pm2.10}$ | $79.71_{\pm1.32}$ | $73.89_{\pm1.12}$ | $67.11_{\pm3.21}$ |
| | | RePHINE | $86.50_{\pm1.78}$ | $79.97_{\pm1.41}$ | $74.05_{\pm1.21}$ | $\mathbf{67.85}_{\pm2.89}$ |
| | SPE | None | $96.32_{\pm0.28}$ | $88.12_{\pm0.41}$ | $80.03_{\pm0.58}$ | $69.64_{\pm0.49}$ |
| | PiPE | VC | $96.57_{\pm0.43}$ | $88.37_{\pm0.82}$ | $80.56_{\pm0.65}$ | $\mathbf{70.92}_{\pm0.92}$ |
| | | RePHINE | $96.87_{\pm0.76}$ | $89.98_{\pm1.05}$ | $80.76_{\pm0.87}$ | $70.46_{\pm0.79}$ |

