# OpenReview forum: "Positional Encoding meets Persistent Homology on Graphs"
_ICML.cc/2025/Conference — ICML 2025 poster_

### Official Review · Reviewer_n4Zk · 2025-03-09

**Overall Recommendation:** 3

**Summary:**

The manuscript introduces a novel positional encoding (PE) methodology for GNNs by combining eigenvalue and eigenvector information from graph Laplacians with persistent homology (PH). The authors present theoretical insights demonstrating that neither PE nor PH independently surpasses the other in expressiveness. A new method termed Persistence-informed Positional Encoding (PiPE) is proposed, theoretically and empirically validated to outperform standalone PE and PH methods in various tasks including molecule property prediction, graph classification, and out-of-distribution generalization.

**Claims And Evidence:**

**W1.** The claim in Section 1 about the drawbacks of partitioning eigenvalue/eigenvector spaces requires clarification. Specifically, the manuscript should clearly state that the basis invariant PEs (e.g. BasisNet proposed by Lim et al. (2023)) can utilize the full eigenvalue/eigenvector space.

**Essential References Not Discussed:**

**W5.** Some relevant references on analyzing the expressive power of PH are missing. For instance, Theorems 1–3 of [1] examine the expressive power of PH in distinguishing non-isomorphic graphs when specific filter functions, such as shortest-path distances, are employed. Additionally, Theorem 4 of [1] investigates the expressive power of PH specifically in distinguishing regular graphs.

[1] Yan et al., Enhancing Graph Representation Learning with Localized Topological Features. JMLR, 2025.

**Experimental Designs Or Analyses:**

**W3.** Empirical validation would benefit from the inclusion of popular benchmarks, such as the OGBG-MOLHIV dataset to provide broader benchmarking and reinforce the generalizability and practical impact of the proposed method.

**W4.** An ablation study explicitly comparing the separate effects of utilizing only 0-dimensional or 1-dimensional persistent diagrams should be conducted. Such an analysis would clarify the relative importance and specific contributions of each dimensional component, enhancing methodological robustness and clarity.

**Methods And Evaluation Criteria:**

Yes

**Other Comments Or Suggestions:**

W6. It would improve readability to index equations explicitly (e.g., The equation on line 307 of page 6).

**Other Strengths And Weaknesses:**

Strengths:

**S1.**	The paper provides a theoretical foundation, elucidating the complementary nature of positional encodings and persistent homology.

**S2.**	PiPE offers demonstrable performance gains across multiple benchmarks and tasks, highlighting practical effectiveness and versatility.

**S3.**	The integration of PH with positional encoding is a novel contribution, expanding the toolset for improving the expressivity of GNNs.

**Questions For Authors:**

N/A.

**Relation To Broader Scientific Literature:**

The manuscript positions itself within the broader context of graph representation learning by directly addressing known limitations in positional encoding methods and persistent homology-based approaches. It builds upon prior works on positional encodings, and complements existing theoretical insights regarding the expressiveness of PH frameworks. By integrating these two methodologies, PiPE advances current understanding and capabilities in distinguishing complex graph structures, particularly within molecular and synthetic benchmarks.

**Theoretical Claims:**

**W2.** A foundational assumption of Proposition 3.1 and subsequent propositions is that the positional encodings should be permutation equivariant and basis invariant, ensuring identical PE outputs for isomorphic graphs regardless of permutation or eigenvector selection. The manuscript must explicitly acknowledge and highlight this critical assumption to reinforce the validity and relevance of the theoretical claims.

---

> ### Author Rebuttal · Authors · 2025-04-01
>
> Many thanks for your thoughtful feedback! We address all your questions and comments below.
>
> > The claim in Section 1 about the drawbacks of partitioning eigenvalue/eigenvector spaces requires clarification.
>
> Thank you for the opportunity to clarify this point. We agree that, in theory, one can utilize the full eigenspectrum of the Laplacian PE. However, in practical scenarios—especially when dealing with datasets containing graphs of varying sizes—the number of eigenmaps, $k$, is treated as a hyperparameter and chosen independently of graph sizes. Most methods, including SPE and BasisNet, follow this approach for network parametrization.
>
> One could, in theory, set $k$ to the maximum graph size in the training set to incorporate the full spectrum. However, this approach becomes memory-intensive and infeasible for large graphs. Moreover, adjustments would still be required during testing if the test set contains graphs larger than those seen during training.
>
> We will add a clarification regarding it in section 1 of the revised version of the paper.
>
> > A foundational assumption....
>
> Thanks for pointing this out. We will highlight this assumption in the revised manuscript.
>
> > Empirical validation....
>
> Based on your feedback, we have performed an additional experiment on OGBG-MOLHIV, and here are the ROC-AUC results over the test dataset for the RW and SPE methods.
>
> | PE method | Diagram         | OGBG-MOLHIV         |
> |-----------|----------------|---------------------|
> | RW        | -              | 0.762 $\pm$ 0.007  |
> |           | PiPE (VC)      | 0.781 $\pm$ 0.005  |
> |           | PiPE (RePHINE) | **0.798** $\pm$ 0.004  |
> | SPE       | -              | 0.776 $\pm$ 0.004  |
> |           | PiPE (VC)      | 0.785 $\pm$ 0.005  |
> |           | PiPE (RePHINE) | **0.791** $\pm$ 0.003  |
>
> > An ablation study explicitly comparing the separate effects of utilizing only 0-dimensional or 1-dimensional persistent diagrams....
>
> Thank you for another excellent suggestion! We conducted an ablation study using $0$-dimensional (VC diagram) and $0$ and $1$-dimensional PH diagrams (RePHINE diagram), with the results presented below. For OGBG-MOLHIV, we report the ROC-AUC, while for ZINC and Alchemy, we report the MAE on the test dataset.
>
> | PE method | Diagram         | OGBG-MOLHIV        | ZINC               | Alchemy            |
> |-----------|----------------|--------------------|--------------------|--------------------|
> | RW        | -              | 0.762 $\pm$ 0.007 | 0.090 $\pm$ 0.005 | 0.121 $\pm$ 0.002 |
> |           | PiPE (VC)      | 0.781 $\pm$ 0.005 | 0.075 $\pm$ 0.011 | 0.113 $\pm$ 0.002 |
> |           | PiPE (RePHINE) | 0.798 $\pm$ 0.004 | 0.070 $\pm$ 0.010 | 0.111 $\pm$ 0.002 |
> | SPE       | -              | 0.776 $\pm$ 0.004 | 0.069 $\pm$ 0.004 | 0.108 $\pm$ 0.001 |
> |           | PiPE (VC)      | 0.785 $\pm$ 0.005 | 0.059 $\pm$ 0.010 | 0.105 $\pm$ 0.003 |
> |           | PiPE (RePHINE) | 0.791 $\pm$ 0.003 | 0.058 $\pm$ 0.007 | 0.103 $\pm$ 0.004 |
>
> > Some relevant references on analyzing the expressive power of PH are missing...
>
> Thank you for providing the reference. We agree that this work is relevant to ours, and we will ensure it is appropriately positioned in the revised version.
>
> > It would improve readability to index equations explicitly (e.g., The equation on line 307 of page 6).
>
>
> Thank you for pointing this out. We will numerate all the equations in the revised version of the paper.
>
> We are grateful for your insightful and constructive feedback. We hope our response has addressed your questions and concerns, and would appreciate your stronger support for this work. Thank you so much!

---

> > ### Comment · Reviewer_n4Zk · 2025-04-04
> >
> > Thank you for your responses.
> >
> > After carefully reviewing the manuscript and all accompanying reviews and rebuttals, I have decided to maintain my score and recommend a acceptance of the paper.

---

> > > ### Author Response · Authors · 2025-04-08
> > >
> > > Thank you for following up and recommending acceptance.
> > >
> > > Could you please let us know if all your concerns have been resolved or if there's anything else you'd like us to address?
> > > Many thanks!

---

### Official Review · Reviewer_rkwW · 2025-03-11

**Overall Recommendation:** 3

**Summary:**

This paper introduces Persistence informed Positional Encoding (PiPE), a novel learnable method that synergizes positional encoding with persistent homology to enhance graph neural networks (GNNs). The authors argue that traditional message-passing GNNs suffer from limitations in expressiveness, being constrained by the 1-Weisfeiler-Lehman test in distinguishing non-isomorphic graphs. They demonstrate that integrating PiPE leads to improved performance in various real-world applications, including graph classification and molecule property predictions, by enabling richer graph representations. The empirical results indicate that PiPE not only outperforms baseline methods in predictive tasks but also shows superior generalizability in out-of-distribution scenarios, showcasing its effectiveness in capturing relevant features for unseen data distributions.

**Claims And Evidence:**

The claims made in the submission are supported by clear evidence.

**Essential References Not Discussed:**

No.

**Experimental Designs Or Analyses:**

The experimental designs are valid.

**Methods And Evaluation Criteria:**

The proposed methods and evaluation criteria make sense.

**Other Comments Or Suggestions:**

No.

**Other Strengths And Weaknesses:**

Strength:
This paper takes a good approach by demonstrating through examples that PH and PE cannot replace each other on their own. Therefore, it makes sense to me that the combination of them leads to better expressivity.

Weakness:
1. The compared baselines are all positional encoding approaches. How about the comparison with PH-based approaches? From my perspective, the only difference between the proposed method and TOGL [1] is that in TOGL the filtration function is applied to node attributes rather than PEs. I wonder how the results might differ.
2. The theoretical analysis is based on unattributed graphs, while the graphs used in experiments are attributed. Besides, the theoretical analysis is about the expressiveness, while the experiments are about regression and classification tasks that are not related to the expressiveness. If the theoretical analysis does not align with the experimental results, what is the significance of the theoretical analysis?

[1] Horn, Max, et al. "Topological graph neural networks." arXiv preprint arXiv:2102.07835 (2021).

**Questions For Authors:**

At the end of the 3rd paragraph in Introduction, the authors mention that topological information may be relevant to downstream tasks. Could you provide some evidence or literature?

**Relation To Broader Scientific Literature:**

The key contributions of the paper are related to graph Transformer.

**Theoretical Claims:**

The theoretical claims in this paper are well-developed.

---

> ### Author Rebuttal · Authors · 2025-04-01
>
> Thank you so much for your thoughtful comments and excellent suggestions! We've acted on all of them, and also address all your concerns, as we describe below.
>
> > The compared baselines are all positional encoding approaches....
>
> Thank you for an excellent suggestion. Based on your feedback, we have conducted an ablation study comparing different PH-based approaches, including vertex-color (VC) filtrations, RePHINE filtrations, and TOGL. Below, we present the test MAE results on the ZINC and Alchemy datasets.
>
> | PE method | Diagram         | ZINC               | Alchemy            |
> |-----------|----------------|--------------------|--------------------|
> | RW        | -              | 0.090 $\pm$ 0.005 | 0.121 $\pm$ 0.002 |
> |           | TOGL           | 0.080 $\pm$ 0.005 | 0.114 $\pm$ 0.002 |
> |           | PiPE (VC)      | 0.075 $\pm$ 0.011 | 0.113 $\pm$ 0.002 |
> |           | PiPE (RePHINE) | **0.070** $\pm$ 0.010 | **0.111** $\pm$ 0.002 |
> | PEG       | -              | 0.187 $\pm$ 0.012 | 0.114 $\pm$ 0.001 |
> |           | TOGL           | 0.143 $\pm$ 0.012 | 0.112 $\pm$ 0.002 |
> |           | PiPE (VC)      | 0.125 $\pm$ 0.017 | 0.112 $\pm$ 0.003 |
> |           | PiPE (RePHINE) | **0.108** $\pm$ 0.022 | **0.111** $\pm$ 0.004 |
> | SPE       | -              | 0.069 $\pm$ 0.004 | 0.108 $\pm$ 0.001 |
> |           | TOGL           | 0.062 $\pm$ 0.003 | 0.112 $\pm$ 0.002 |
> |           | PiPE (VC)      | 0.059 $\pm$ 0.010 | 0.105 $\pm$ 0.003 |
> |           | PiPE (RePHINE) | **0.058** $\pm$ 0.007 | **0.103** $\pm$ 0.004 |
>
> > The theoretical analysis is based on unattributed graphs....
>
> Thank you for another insightful remark. Based on your feedback, we have now conducted an empirical study to evaluate the expressiveness of standard PH, PH + LPE and PiPE on the BREC[1] dataset consisting of unattributed graphs across five categories: Basic, Regular, Extension, CFI, and  Distance. The table below reports accuracy, with the numbers in parentheses indicating the number of graph pairs in each dataset.
>
> | Dataset          | PH   | PH+LPE | PiPE  |
> |-----------------|------|--------|-------|
> | Basic (60)      | 0.03 | 0.10   | **0.72** |
> | Regular (50)    | 0.00 | 0.15   | **0.40** |
> | Extension (100) | 0.07 | 0.13   | **0.67** |
> | CFI (100)       | 0.03 | 0.03   | 0.03  |
> | Distance (20)   | 0.00 | 0.00   | **0.05** |
>
> [1] Y. Wang et al. An empirical study of realized GNN expressiveness, ICML 2024.
>
> > At the end of the 3rd paragraph in Introduction,...
>
> Several recent works provide evidence that topological information is relevant for downstream tasks in machine learning.
>
> For instance, [1] highlights that incorporating topological structures enhances relational learning by capturing higher-order dependencies beyond standard graph-based representations. Additionally, [2] extends topological networks by incorporating persistence and equivariance, demonstrating applications in molecular dynamics, drug property prediction, etc. Moreover, [3] introduces $E(n)$-equivariant topological neural networks, showing their effectiveness in geometric and molecular modeling tasks. Similarly, [4] proposes TopoDiffusionNet, which integrates topological priors into diffusion models, leading to improved performance in generative modeling.
>
> These works collectively support the claim that topological information can benefit various downstream applications. We will add these references in the revised version of the manuscript.
>
> [1] T. Papamarkou et al. Position: Topological Deep Learning is the New Frontier for Relational Learning, ICML 2024
>
> [2] Y. Verma et al. Topological Neural Networks go Persistent, Equivariant, and Continuous, ICML 2024
>
> [3] C. Battiloro et al. $E(n)$ Equivariant Topological Neural Networks, ICLR 2025
>
> [4] S. Gupta et al. TopoDiffusionNet : A topology aware diffusion model, ICLR 2025
>
> Many thanks again for your constructive feedback and incisive comments. We hope we have satisfactorily addressed your concerns, and would appreciate the same being reflected in an increased score.

---

> > ### Comment · Reviewer_rkwW · 2025-04-05
> >
> > I thank the authors for their efforts to address my concerns. I have no further questions.

---

> > > ### Author Response · Authors · 2025-04-08
> > >
> > > Thank you for following up! We are glad that all your concerns have been addressed. Could you therefore kindly consider upgrading your score accordingly from the initial score of 3?
> > >
> > > We believe the additional evidence based on your feedback has reinforced the merits of this work, and will be sure to incorporate it in the updated version.  Many thanks!

---

### Official Review · Reviewer_VeW5 · 2025-03-13

**Overall Recommendation:** 4

**Summary:**

The paper proposes a novel graph neural network architecture combining existing positional encoding (PE) strategies with persistent homology (PH) and shows the resulting method is strictly more expressive than previous works using either PE or PH.

**Claims And Evidence:**

The theoretical claims seems to be supported by theoretical proofs and convincing experimental results.

**Essential References Not Discussed:**

N/A

**Experimental Designs Or Analyses:**

Different variations of the proposed method obtained by combining different PE strategies and different PH vectorization methods are evaluated on different graph datasets, consistently showing the improvements obtained by combining these strategies.

**Methods And Evaluation Criteria:**

See below.

**Other Comments Or Suggestions:**

- could you numerate all the equations? That is very helpful for the reader and the reviewers in order to refer to them

- Proposition 3.1:  what is the multiset of Laplacian positional encodings (LPE)? Can you define this more precisely?

- Proposition 3.1:  is S2 and S3 true for any k?

- Proposition 3.1:  what about S2 and S3 together? i.e. if b_0 and b_1 are both unequal, is it possible the eigenvectors are still equal?

- Sec 4. page 5, what is A in the equations? are z_v scalars?

- Sec 4. page 5, how are the diagrams turned into node or edge features? How are $\Psi_i^l$ functions defined? This is briefly mentioned in the first paragraph in the next page but 1) it's not defined and 2) it would be beneficial to briefly define all variables in the equations when they are stated, to avoid confusing the reader. Because this step seems particularly important for the method, I think the authors could reserve a short paragraph to explain how this vectorization and bijection work.

**Other Strengths And Weaknesses:**

The paper proposes and efficient and powerful method to strictly improve the expressivity of graph neural networks.
This seems a relevant paper for the scientific community.

I think the main weakness of the manuscript is that it is at times unclear. See comments below.
Improving clarify could help the adoption of the proposed method.

**Questions For Authors:**

- Proposition 4.2: is the point of this proposition that computing PH at each layer is more expressive than doing it only once? Do I understand correctly that PH + LPE is essentially similar to a 1-layer PiPE to computed features used by a generic predictor?

**Relation To Broader Scientific Literature:**

The manuscript is well motivated and provides context in the related literature.
I am quite familiar with graph deep learning and persistent homology / topological methods in this context, but it's not my main expertise, so I might be missing some related works.

**Theoretical Claims:**

I didn't verify the theoretical proofs.

---

> ### Author Rebuttal · Authors · 2025-04-01
>
> Many thanks for your thoughtful review. We address your concerns and incorporate your suggestions below.
>
> > could you numerate all the equations? That is very helpful for the reader and the reviewers in order to refer to them
>
> Thank you for pointing this out. We will numerate all the equations in the revised version of the paper.
>
> >Proposition 3.1: what is the multiset of Laplacian positional encodings (LPE)? Can you define this more precisely?
>
> The multiset of Laplacian positional encodings consists of the PE vectors (eigenmaps) assigned to all nodes, derived from the Laplacian eigen-decomposition of the graph structure. We will be sure to make it clear in the revised version of the paper.
>
> > Proposition 3.1: is S2 and S3 true for any k?
>
> Yes. S2 and S3 are true for any $k$. Specifically, for any given $k$, we can always find graphs for which S2 holds. Similarly for S3.
>
> > Proposition 3.1: what about S2 and S3 together? i.e. if $\beta_0$ and $\beta_1$ are both unequal, is it possible the eigenvectors are still equal?
>
> Thanks for an interesting question. Yes. For any $k$, there exist graphs $G,G'$ such that $\beta_0(G) \neq \beta_0(G')$ and $\beta_1(G) \neq \beta_1(G')$, but $\Phi_{k}(G) = \Phi_{k}(G')$, where $\Phi$ denotes the laplacian positional encoding based on $k$-lowest eigenmaps. We describe the proof below.
>
> Let $K_i$ denote the complete graph with $i$ nodes. Consider a graph $G= \cup_{i=1}^{n/2}K_1 \cup K_3 $ --- here $K_1 \cup K_3$ denotes a graph with two components comprising one isolated node and a triangle i.e. $\beta_0(G) = n, \beta_1(G) = n/2 $. Also, consider $G'= \cup_{i=1}^{n/2} (K_1 \cup K_1 \cup K_1 \cup K_1)$ --- i.e., $4n/2$ isolated nodes i.e. $\beta_0(G') = 2n, \beta_1(G') = 0 $. The $k$ smallest eigenvalues corresponding to $G$ are all equal to 0 with the identical constant eigenvector, when $k \leq n$.  Similarly,  $G'$ has the same eigenvalues with identical constant eigenvectors.  However, the number of connected components and basis cycles in $G$ and $G'$ differ, i.e. $\beta_1(G) \neq \beta_1(G')$ and $\beta_0(G) \neq \beta_0(G')$.
>
> > Sec 4. page 5, what is A in the equations? are $z_v$ scalars?
>
> Thanks for pointing this out. The $z_{v}$ are the latent encodings obtained at layer $\ell$ after GNN message passing, and are vectors. $A$ represents the graph structure i.e. the adjacency matrix which is used to compute the $0$-dim and $1$-dim PH diagrams.
>
> > Sec 4. page 5, how are the diagrams turned into node or edge features?
>
> Thank you for the opportunity to elaborate on this. To obtain node features from $\mathcal{D}_0$ we note that $|\mathcal{D}_0|=n$, and, therefore, we can define a bijection between $V$ and $\mathcal{D}_0$. In particular, we can associate the death of a connected component with a node and leverage that fact to obtain node features from $\mathcal{D}_0$. For dimension 1, although $|\mathcal{D}_1|$ is equal to the number of independent cycles, we can use dummy tuples for edges that are not linked to any cycle. Details of this procedure can be found in Appendix A.4 in [1]. We agree this is important for the method and we will add a paragraph to clarify the diagram vectorization in the revised manuscript.
>
> [1] M. Horn et al. Topological Graph Neural Networks, ICLR, 2022.
>
> > Proposition 4.2: is the point of this proposition that computing PH at each layer is more expressive than doing it only once?
>
> Thank you for the opportunity to underscore the significance of Proposition 4.2. In Proposition 4.2, the idea is to show the message-passing layers of PiPE brings in additional expressivity over simply applying PH on top of base positional encoders. Thus, this result motivates the adoption of message-passing in PiPE.
>
> > Do I understand correctly that PH + LPE is essentially similar to a 1-layer PiPE to computed features used by a generic predictor?
>
> For the sake of the analysis, PH + LPE represents the $0$-th and $1$-th persistence diagrams from a filtration function $f$ on the LPE positional encoder $p_v$ for all $v$. As you pointed out, this is similar to a $0$-layer PiPE (i.e., before message passing) --- $f(p_v)$ is the generic predictor that you mentioned (e.g., an MLP mapping $p_v$ to a real number).
>
> We are grateful for your insightful questions that have helped shed light on some subtle and salient aspects of this work. Thank you for your support!

---

### Official Review · Reviewer_G9xp · 2025-03-14

**Overall Recommendation:** 4

**Summary:**

This paper introduces a novel method—Persistence informed by Positional Encoding (PiPE)—which unifies positional encoding (PE) with persistent homology to enhance the expressivity of graph neural networks . The authors provide both theoretical insights and empirical evidence, demonstrating that PiPE overcomes some of the limitations inherent in standalone PE or PH approaches. The method is evaluated on diverse tasks and the results are quite promising.


## update after rebuttal

**Claims And Evidence:**

The detailed theoretical analysis—including propositions, lemmas, and discussions around the expressive power of the method within the k-WL hierarchy—provides strong justification for the proposed approach. This solid theoretical grounding makes the work a valuable contribution to the graph learning literature.

**Essential References Not Discussed:**

Related work is sufficiently discussed.

**Experimental Designs Or Analyses:**

The experimental design is rigorous, incorporating a diverse set of real-world benchmarks and controlled synthetic tasks that thoroughly assess both the performance and generalizability of the proposed method. Moreover, detailed ablation studies and runtime analyses substantiate its scalability and performance improvements, reinforcing the empirical evidence supporting the approach.

**Methods And Evaluation Criteria:**

The paper leverages a broad range of benchmarks—including both real-world and synthetic networks—alongside robust theoretical contributions, which add value to paper and help to convey the message

**Other Comments Or Suggestions:**

The paper could benefit from further clarification or additional examples to help readers less familiar with advanced topological concepts.

**Other Strengths And Weaknesses:**

The paper introduces a novel approach that creatively combines positional encoding with persistent homology, offering a fresh perspective on enhancing GNN expressivity. Its contributions are further solidified by a rigorous theoretical analysis that thoroughly underpins the proposed method. However, the approach comes with increased complexity and is currently limited to 1-dimensional simplicial complexes.

**Questions For Authors:**

Can the method be trivially extended to higher dimensional simplicial complexes?

**Relation To Broader Scientific Literature:**

The paper has nice contributions to the graph representation learning literature and could be tied with works around how graph neural networks capture both structural and feature-based information [1].

[1] What Do GNNs Actually Learn? Towards Understanding their Representations, Nikolentzos et al.

**Theoretical Claims:**

The theoretical claims included in the paper are robust and well-founded, though there remains a slight possibility I might have overlooked something.

---

> ### Author Rebuttal · Authors · 2025-04-01
>
> Many thanks for your constructive feedback, and for appreciating the different aspects of this work. We address your comments below.
>
> > The paper could benefit from further clarification or additional examples to help readers less familiar with advanced topological concepts.
>
> Thank you for an excellent suggestion. We will add a detailed background section in the appendix in the updated version.
>
> > Can the method be trivially extended to higher dimensional simplicial complexes?
>
> Thank you for an interesting question. Two steps are required to extend the proposed method to higher dimensional simplicial complexes. First, the positional encodings for various higher-order simplical complexes can be computed using the Hodge Laplacian [1, 2], which generalizes graph Laplacian and serves as a higher dimensional shift operator. Second, one needs to compute the persistent homology (PH) diagram over higher-order simplices, e.g., see [3]. Additionally, one could replace the GNN with a persistence-encoding Topological Neural Network (TNN), e.g. [4], integrating positional encodings and enabling message passing over simplices for downstream tasks.
>
> [1] L.-H. Lim. Hodge Laplacians on graphs, SIAM Review 62.3 (2020): 685-715.
>
> [2] Z. Yan et al. Cycle Invariant Positional Encoding for Graph Representation Learning,  LoG, 2023.
>
> [3] F. Chazal et al. Convergence Rates for Persistence Diagram Estimation in Topological Data Analysis, JMLR 16 (2015): 3603-3635.
>
> [4] Y. Verma et al. Topological Neural Networks go Persistent, Equivariant, and Continuous, ICML, 2024.
>
> We are grateful for your thoughtful feedback and suggestions. Thank you so much!

---

> > ### Comment · Reviewer_G9xp · 2025-04-03
> >
> > I would like to thank the authors for their response.
> >
> > I would appreciate if the discussion around high simplicial complexes would be introduced in the future work/discussion of the paper, as it holds value to the graph representation learning literature.
> >
> > I will maintain my score and I would like to congratulate the authors for the very nice paper.

---

> > > ### Author Response · Authors · 2025-04-08
> > >
> > > Thank you for the follow-up. We'll be sure to include the discussion around simplicial complexes as you kindly suggested.  We're grateful for your thoughtful feedback and your strong support for this work. Many thanks!

---

### Decision · Program_Chairs · 2025-05-01

**Decision:**

Accept (poster)

**Comment:**

This submission presents a novel framework for imbuing positional encodings of graphs with topology-based features (as measured using persistent homology). Next to providing the theoretical properties, i.e., advantages and shortcomings, of both approaches, the manuscript contains extensive experiments in a variety of tasks. Reviewers were unanimous in their positive assessment of the paper, and the strong rebuttal helped in alleviate some of the concerns. Of particular noteworthiness to the AC is the high quality of the writing and the illustrations of the paper. Already in its current state, this submission is clearly ready for presentation at the conference and will prove to be a relevant work in graph representation learning.

I trust the authors to implement the changes they promised during the rebuttal in the camera-ready version. Along those lines, I would also like to kindly request authors to include tables of experimental results into the appendix whenever they deem it appropriate. These could nicely complement the graphical depictions of experiments in the paper.